


# Energy, water and carbon exchanges in managed forest ecosystems : description, sensitivity analysis and evaluation of the INRAE GO+ model, version 3.0.

Virginie Moreaux[1,11], Simon Martel[1,10], Alexandre Bosc[1], Delphine Picart[1], David Achat[1], Christophe Moisy[1], Raphael Aussenac[1], Christophe Chipeaux[1], Jean-Marc Bonnefond[1], Pierre Trichet[1], Rémi Vezy[2], Vincent Badeau[3], Bernard Longdoz[3], André Granier[3], Olivier Roupsard[4], Manuel Nicolas[5], Kim Pilegaard[6], Giorgio Matteucci[7], Claudy Jolivet[8], Andrew T. Black[9], Olivier Picard[10], and Denis Loustau[1]

[1]Bordeaux-Sciences-Agro, INRAE, UMR ISPA, Villenave d'Ornon, F-33140, France
[2]AMAP, Univ Montpellier, CIRAD, CNRS, INRAE, IRD, Montpellier, France
[3]INRAE, UMR SILVA 1434, Champenoux, F-54280, France
[4]Eco&Sols, Univ. Montpellier, CIRAD, INRAE, IRD, Montpellier SupAgro, Montpellier, France
[5]ONF, Département RDI, Fontainebleau, F-77300, France
[6]Technical University of Denmark, Department of Environmental Engineering, 2800 Kgs. Lyngby, Denmark
[7]CNR,ISAFOM, I-80056 Ercolano (NA) Italy
[8]INRAE, US1106 InfoSol, Orléans, F-45075, France
[9]Faculty of Land and Food Systems, Vancouver, BC - V6T 1Z4, Canada
[10]CNPF-IDF, Paris, F-75000 France
[11]Université de Grenoble-Alpes,CNRS, UMR IGE 5001, Grenoble, F-38058, France

**Correspondence:** Denis Loustau (denis.loustau@inrae.fr)

**Abstract.** The mechanistic model GO+ describes the functioning and growth of managed forests based upon biophysical and biogeochemical processes. The biophysical and biogeochemical processes included are modelled using standard formulations of radiative transfer, convective heat exchange, evapotranspiration, photosynthesis, respiration, plant phenology, growth and mortality, biomass nutrient content, and soil carbon dynamics. The forest ecosystem is modelled as three layers, namely the

tree overstorey, understorey and soil. The vegetation layers include stems, branches and foliage and are partitioned dynamically between sunlit and shaded fractions. The soil carbon sub-model is an adaption of the Roth-C model to simulate the impact of forest operations. The model runs at an hourly time-step. It represents a forest stand covering typically 1 ha and can be straightforwardly up-scaled across gridded data at regional, country or continental levels. GO+ accounts for both the immediate and long-term impacts of forest operations on energy, water and carbon exchanges within the soil-vegetation-atmosphere contin-

uum. It includes exhaustive and versatile descriptions of management operations (soil preparation, regeneration, vegetation control, selective thinning, clear-cutting, coppicing, etc.), thus permitting the effects of a wide variety of forest management strategies to be estimated: from close-to-nature to intensive. This paper examines the sensitivity of the model to its main parameters and estimates how errors in parameter values are propagated into the predicted values of its main output variables. We show how the model performs when compared with observations such as time series of forest-atmosphere exchanges of energy,

water and CO2 monitored over Douglas fir, European beech and pine forests of different ages as well as long-term series of





tree growth, soil water and soil carbon data recorded at continuously monitored forests plots. We also illustrate the capacity of the GO+ model to simulate the provision of key ecosystem services, such as the long-term storage of carbon in biomass and soil under various management and climate scenarios.

## 1 Introduction

Carbon sequestration by forest ecosystems offsets a significant part of the global carbon emissions from burning fossil fuel (Pan et al. 2011). Forests are assumed to have the potential to be a low-cost effective measure for keeping the global temperature increase below +2°C (Griscom et al. 2017). Hence, the conversion of other land-use types into forests and the management of existing forests have been included in the portfolio of environmental actions to allow compliance with the international agreements proposed at Kyoto, Paris and subsequent conferences (Grassi et al. 2017). The enhanced management of existing forests and new plantations may play a substantial role in attenuating the increase in atmospheric CO2 concentration; especially for forests in temperate Europe and Russia (Bright et al. 2017). Managed forests also constitute the major source of material for wood-derived products. The growth of the world's human population is creating an increasing demand for such wood and fibre products; this demand is also leading to pressure to intensify the management of forests. Indeed, 22% of global ice-free land is covered by forests subject to diverse management strategies for wood production and other services; this compares with 9% occupied by unmanaged forests (IPCC 2019 report). In Europe, 86% of the forested area is managed, although with a large range of intensity. These numbers show that the dynamics of more than two-thirds of the world's forest are dominated by human activities.

In this context, the impacts of the management of European forests on climate are a matter of debate. The biophysical impacts on climate through, e.g., heat and radiation exchanges, and the biogeochemical role of forests, e.g., carbon sequestration, may be antagonistic and could cancel out (Bright et al. 2012, 2017, Luyssaert et al. 2018). In addition, the climate impacts of forest management at local, regional and global scales are diverse. Management affects the entire forest life cycle through many aspects such as the soil preparation, drainage, fertilization, tree stand species composition, age-class distribution, tree regeneration, thinning and harvest, control of diseases, pests and fires, land-use changes, etc. Many forest operations involved in modern forestry drastically change key canopy properties such as its albedo, roughness, leaf area index, standing biomass, and number of stems per hectare (Garcia et al. 2014, Kuusinen et al. 2014, Otto et al. 2014). Important soil properties (heat and water storage capacities, cation exchange capacity, nutrient stocks) are also affected by forest operations that are common in managed forests (logging, soil preparation, drainage, fertilization, liming) with significant but controversial impacts on carbon dynamics (Stromgren et al. 2013, Achat et al. 2015, Jurevics et al. 2016, Erb et al. 2017, Zhang et al. 2018). The forest understorey is also targeted by management practices aimed at decreasing the competition between the trees and understorey vegetation, or reducing the stands' vulnerability to fire (Borys et al. 2016).





Furthermore, the environmental effects of forest management and land-use changes have long been shown to interact with local climate conditions and forest characteristics such as albedo and roughness. Both climate models and observations have shown that the expansion of forests has some contrasting effects in boreal regions: there climate benefits would be limited or negative, as compared with the tropics, where enhanced evapotranspiration from forested areas reduces the sensible heat flux and may

enhance cloud formation at regional scale (Cutrim et al. 1995, Betts, 2000, Lee et al. 2011, Bala et al. 2007, IPCC Report 2019) . The aridity also plays a key role, giving forests a net effect of slightly warming arid zones, due to the overwhelming impact of enhanced net radiation; in contrast a net cooling effect would result from afforestation or reforestation of humid zones due to enhanced latent heat flux (Huang et al. 2018). Climatic impacts of forest also depend on the tree species, in particular their specific albedo and evapotranspiration (Naudts et al. 2016, Ahlswede and Thomas, 2017). Through changes in albedo and in

convective heat exchanges with the lower troposphere, forest management may impact the surface and planetary boundary layer temperature by the same magnitude as that from land-use changes (Bright et al. 2012, Luyssaert et al. 2014, Ahlswede and Thomas 2017). However, quantifying these biophysical impacts on climate is a complex procedure and therefore not accounted for in impact studies (Yousefpour et al. 2018) — as a result they have so far been ignored in climate treaties.

The forest products harvested from managed forests are also accounted for under a controversial "substitution" effect, that is,

the replacement of emissions-intensive materials by wood products; a process that reduces emissions in other sectors (IPCC report 2019). This putative substitution effect is difficult to quantify due to the large diversity of wood products, transportation and transformation processes and product life cycles. Indeed, the substitution coefficient, the ratio of fossil carbon avoided to the bio-sourced carbon used, has been found to vary from -2.0 to 15 (Sathre and O'Connor, 2010). Nevertheless, considering the impact of wood products on the emissions of fossil carbon is essential when assessing and comparing the climate impacts of

forest management strategies (Schlamadinger and Marland 1996). It should be accounted for in forest models. Including such an effect in impact studies implies that forest growth models must be connected to wood product life cycles, and, among others, to details of how the carbon is apportioned to the different products harvested and of their temporal dynamics (Pichancourt et al. 2018).

      Mechanistic, process-based models of forest biophysics and biogeochemistry display a range of ability at representing forest

management effects; their ability depends on their temporal and spatial resolution, and on the level to which they have been simplified. The most detailed dynamic stand-scale models, designed for describing a forest patch of typically one hectare area, include operations such as thinning and harvest (Deckmyn et al. 2008, Gutsch et al. 2011, Guillemot et al. 2014), and their frequency and intensity. They also allow the modeller to select the trees to be cut and harvested (Lindner et al. 1997). However, most models restrict the selection of the tree parts harvested to the stem and ignore the impacts on soil carbon of the removal

of other elements such as branches, foliage or stumps. Until recently, global vegetation models have prioritized their efforts on the effects of land-use changes and tend to oversimplify the impacts of the management, that are reduced to age-class and functional type distributions (Bellassen et al. 2010 a and b, Harper et al. 2018, but see the implementation of management schemes across Europe by the model ORCHIDEE- CAN by Luyssaert et al. 2018). A few models, e.g., Rasche et al. (2013), do account for the size distribution of the harvested stems which allows one to realistically route the raw harvest products among

energy, pulp, fibre, industrial uses, plywood and panels, and other building material (Schlamadinger and Marland 1996, Masera



et al. 2003).

To our knowledge no process-based model, local, regional or global, accounts for the effects of soil preparation techniques and understorey management on the energy balance, canopy properties, and ecosystem water and carbon balances. A few models can be coupled with other models of product life cycles, paving the way for assessing the impacts of the entire forest product

life cycle. Models based on forest inventory data, so-called data models, and empirical growth and yield models may represent accurately the management effects on tree growth and wood production (Karjalainen et al., 2003, Kurz et al. 2009, Pilli et al. 2017). However, they do not account for the impact of climate and biogeochemical processes, nor do they allow new management strategies to be implemented. These models are not designed for simulating ecosystem functions — essentially they model growth and production under steady environmental conditions.

To progress our understanding of the role and functions of managed forests and their behaviour in a rapidly changing world, we present a mechanistic, process-based model called GO+. The model simulates the functioning and growth of temperate managed forests. GO+ accounts for both the immediate and long-term effects of forest operations on energy, water and carbon exchanges within the soil-vegetation-atmosphere continuum. It predicts the temporal dynamics of the above-ground and below-ground biomass of standing and harvested trees, ground surface vegetation and soil. The model is designed to be applied

at large scale, i.e., over typically 10,000 grid points and 150 years. It has therefore been developed considering the trade-off between the need for a realistic prediction of individual tree growth, forest production and ecosystem functions at the country and regional levels, and the representation of the main biogeochemical and biophysical processes required for ensuring its robustness under climate and management scenario combinations. GO+ includes a comprehensive and versatile description of management operations (soil preparation, regeneration, vegetation control, selective thinning, clear-cutting, coppicing, etc.)

allowing a wide variety of forest management strategies to be accounted for, from close-to-nature to intensive. In what follows, we first describe, the suite of processes implemented in GO+ from the radiation balance of the plant canopy to growth, phenology and mortality of individual stems. The parameterization and verification of the model is then presented. We examine the sensitivity of the model to its main parameters and to the driving climate variables. From the results of this analysis, we estimate how errors in parameter values are propagated into the main output variables. Finally, we show how the model

performs through comparisons with different sets of observations such as temporal series of forest-atmosphere exchanges of energy, water and $CO_2$ monitored over Douglas fir, European beech and maritime pine forests (Pseudotsuga Menziesii, Fagus sylvatica and Pinus pinaster, respectively) of different ages, and long time series of tree growth, soil water and soil carbon data recorded at permanent forest plots.

## 2    Model description

This section describes version 3.0 of the model GO+. The model has been developed in parallel to a series of experimental and theoretical developments which were formalized in preliminary versions of the model (Loustau et al. 2005, Ciais et al. 2011). The model is primarily aimed at simulating managed forest stands and has been applied to various species (eucalyptus, Douglas fir, European beech, maritime pine) and management schemes (standard, coppice, self-thinning). In the interests of





t

**Table 1.** List of the forcing meteorological variables driving the GO+ v3.0 model.

| Symbol | Description | Entity (1) | Unit |
|---|---|---|---|
| $CO_2$ | Air $CO_2$ concentration | Air | mol $CO_2$ mol air $^{-1}$ |
| $e_w$ | Air water vapor pressure | Air | Pa |
| $LW\downarrow$ | Downward flux density of longwave radiation | Atmosphere | W m$^{-2}$ |
| $O_2$ | Air $O_2$ concentration | Air | mol $O_2$ mol air $^{-1}$ |
| $P$ | Atmospheric pressure | Atmosphere | Pa |
| $Rain$ | Gross precipitation | Atmosphere | kg $H_2O$ m$^{-2}$ hr$^{-1}$ |
| $SW\downarrow$ | Downward flux density of shortwave radiation | Atmosphere | W m$^{-2}$ |
| $T_a$ | Air temperature | Air | °C |
| $Tr_{soil}$ | Soil reference temperature | Soil | °C |
| $U_{ref}$ | Horizontal wind speed | Air | m s$^{-1}$ |
| $\beta$ | Solar elevation angle | Sun | radians |
| $\delta e_w$ | Air vapor pressure saturation deficit | Air | Pa |

brevity, most of the equations and submodels already published in the literature are reported in the supplementary material; here we present only the main adaptations and innovations of the model.

## 2.1 Overview

The model runs on an hourly time-step and is forced by meteorological variables (Table 1). It describes the energy balance, bio-
5 geochemical functioning and the development, growth and mortality of trees. The complete list of model prognostic variables together with their symbols and units is provided in Appendix A. The model parameters are presented in the supplementary material, Table S1. The vegetation is represented by a two-layer canopy corresponding to the trees and ground vegetation (Fig. 1). The core model includes the main biophysical and geochemical processes of the energy, water and carbon balances, and simulates dynamically the plant growth in height, leaf area, biomass and stem diameter, as well as vegetation dynamics
(phenology, regeneration, senescence and mortality induced by ecological events or management). The upper canopy layer is conceived as a collection of individual trees composed of foliage, branches, stemwood, bark, stump, taproot, coarse roots, small roots and fine roots. The ground vegetation is a simple homogeneous layer including three parts: foliage, roots and a perennial part that corresponds to either rhizomes, seeds or the woody parts of understorey species.

The model calculations start from solving the aerodynamic and radiation transfers, energy balance and water cycle and end
with the resolution of carbon processes, plant growth and mortality. It includes several feed back processes (not shown in Fig.1 for clarity) namely the effects of soil water and carbon content on vegetation layers, the canopy feedback of the atmospheric exchanges of radiation and wind speed. The competition for light resource between the tree and understorey layers is explicit whereas the two entities are treated equally for the access to the soil water resource. For allowing GO+ to be run over large spa-




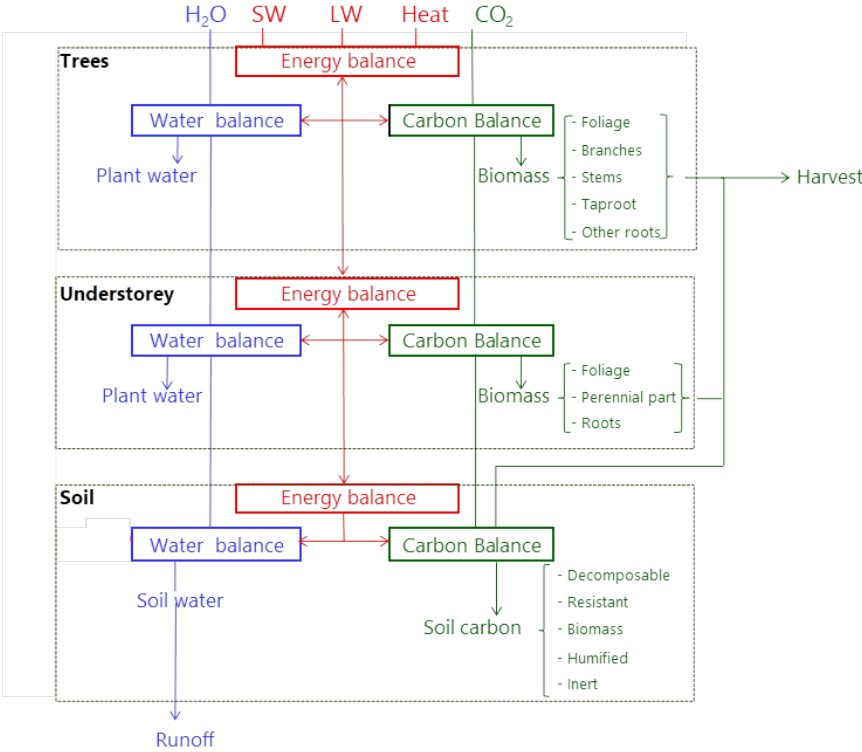

**Figure 1.** Overview of the GO+ model. The carbon exported as harvested material can be composed of stems, branches, foliage, stumps and coarse roots. The carbon flow from tree and understorey into the soil includes litter from foliage, branches and roots as well as harvest residue and dead parts of the understorey.

tial and temporal domains with sufficient resolution, the 3.0 version of GO+ used two main simplifying assumptions releasing model calculations from time consuming iterative computations as follows. First, the feedbacks of canopy sources on the air temperature, humidity and $CO_2$ concentration are neglected, implying that the profile of scalar concentrations gradients within the canopy are not accounted for. Second, some simple analytical solutions of the radiation transfer and energy balance calcu-

5    lations are used instead of iterative calculations, which implies a limited number of approximations detailed in the description section.

## 2.2 Radiation transfer

Each canopy layer is treated as an isothermal turbid medium where intercepting elements, the foliage and above-ground woody parts, are distributed uniformly or clumped. The calculation is operated for each layer from the top to the bottom layer. The

10    transfer of direct and diffuse shortwave radiation, $SW$, and long-wave radiation, $LW$, through each canopy layer is calculated using the Beer-Lambert law of light attenuation with a second order scattering. In the shortwave domain, the GO+ model follows the approach described by de Pury and Farquhar (1997) with few adaptations described below.





### 2.2.1 Foliage

For both the trees and understorey, GO+ allows a dynamic partitioning between sun and shade components for each canopy layer (Eq. S1). The canopy reflection coefficients for diffuse and direct beam irradiance are calculated from the leaf optical characteristics (reflectance, transmittance and absorbance), the diffuse and direct canopy extinction coefficients, $k_{d,c}$ and $k_{b,c}$ 

and, for the later, solar elevation (Eqs S2-S4). The extinction coefficients $k'_{d,c}$ and $k'_{b,c}$, where the primes indicate scattered radiation, are then used to determine the fractions of light absorbed and scattered by the sunlit and shaded parts of the foliage, thus accounting for the second order scattering of shortwave radiation within each canopy layer. The shortwave radiation absorbed by the sunlit and shaded fractions of each canopy layer is given by the sum of direct, diffuse and scattered-beam components (Eq. S5-S7). The absorption of the longwave radiation intercepted by each canopy layer is also simulated using

the isothermal turbid medium analogy and Beer-Lambert's law as detailed in Eq. S8-S9.

### 2.2.2 Woody parts of the tree canopy

The same formalism used for the foliage is used for modelling the passage of both shortwave and longwave radiation through non-leafy parts of the canopy, i.e. the tree branches and stems. The tree canopy leaf area index, $LAI_T$, is substituted by the wood area index, $WAI_T$, the latter being calculated from the above-ground biomass, mean canopy height, stem density per

hectare, mean stem diameter and a trunk shape factor (Eqs. S10).

### 2.3 Energy balance

The exchanges of long-wave radiation between soil, canopy and the atmosphere are calculated according to the analytical solution proposed by Jones (1992) with minor adaptations as follows. First, for each layer $c$, the net isothermal radiation, $Rn_{i,c}$ is calculated from the $SW$ and $LW$ radiative balance, assuming that leaf and air temperature are equal.

$$Rn_{i,c} = SW_{a,c} + LW_{a,c} - 2 \times K_{LW,c} \times \epsilon \times \sigma \times Ta_z^4 \tag{1}$$

where $K_{LWc}$, the emission coefficient for thermal radiation by the canopy layer is calculated following the model by Berbigier and Bonnefond (1995) completed with a term for thermal radiation from the leafless parts of the canopy (stem and branches) as :

$$K_{LWc} = 1 - \exp(k_{LW1} \times LAI_c + k_{LW2} \times LAI_c^2 - k_{d,c,w} \times WAI_c) \tag{2}$$

where $k_{LW1,c}, k_{LW2,c}$ are extinction coefficients of foliage and $k_{d,c,w}$ the extinction coefficient of woody parts for diffuse radiation.

The longwave radiation and heat transfer are calculated using a resistance analogue scheme with a combined resistance to heat





transfer, $r_{HR,c}$, that is calculated from the resistances to convective and radiative transfer, $r_{H,c}$ and $r_{R,c}$ respectively:

$$r_{H,c} = \frac{U_z}{u*_c^2} \tag{3}$$

$$r_{R,c} = \frac{\rho_a \times c_p}{2 \times 4 \times K_{LWc} \times \sigma \times \epsilon \times Ta_z^3} \tag{4}$$

$$r_{HR,c} = \frac{1}{\frac{1}{r_{H,c}} + \frac{1}{r_{R,c}}} \tag{5}$$

Last, the temperature of each canopy layer and air, $Ts_c$, is derived by combining radiative and convective transfers :

$$Ts_c = Ta_z + \frac{Rn_{i,c} \times \dfrac{\gamma}{g_{ctot,c}} \times r_{HR,c}}{\rho_a \times c_p \times (\dfrac{\gamma}{g_{tot,c}} + s \times r_{HR,c})} - \frac{r_{HR,c} \times \delta e_w}{(\dfrac{\gamma}{g_{tot,c}} + s \times r_{HR,c})} \tag{6}$$

Long-wave emission and net radiation absorbed are then given for each canopy layers by Eqs. 7-8:

$$LW_{e,c} = 2 \times K_{LW,c} \times \epsilon \times \sigma \times Ts_c^4 \tag{7}$$

$$Rn_c = SW_{a,c} + LW_{a,c} - 2 \times K_{LW,c} \times \epsilon \times \sigma \times Ts_c^4 \tag{8}$$

The changes in storage of heat into the above-ground biomass, and air and water vapour within the canopy layer is neglected. The soil heat flux, $G$, is:

$$G = \frac{h}{z_{ref}} \times (Ts_{soil} - Tref_{soil}) \tag{9}$$

where $h$ is the thermal conductivity of soil between the reference depth and the top layer of soil in contact with the atmosphere, $Ts_{soil}$ the soil surface temperature and $Tref_{soil}$ the temperature at the lower soil limit taken as the mean annual temperature

of the site.

## 2.4  Momentum and heat transfer

The fluxes of sensible and latent heat from each canopy layer and the soil into the atmosphere at the reference level $z$ are formalized as a transfer through two resistances in series :

- the aerodynamic resistance to momentum transfer under neutral conditions, (related stability parameters equal zero),

$r_{H,c}$ (Eq. 3) is related to the canopy layer height, $h_c$, stem density, $SD_c$, and leaf area index, $LAI_c$, calculated according to the formulation proposed by Nakai et al. (2008):

$$u*_c = U_{ref} \times k \times (\log \frac{z_{ref} - d_c}{z_{0c}})^{-1} \tag{10}$$

with the wind speed at a reference height, $U_{ref}$, is derived from values provided by meteorological data using a logarithmic attenuation profile. The roughness length, $z_{0c}$, and displacement height, $d_c$, are modelled as follows:

$$d_c = [1 - \frac{1 - \exp(-k_1 \times SD_c)}{k_1 \times SD_c} \times \frac{1 - \exp(-k_2 \times LAI_c)}{k_2 \times LAI_c}] \times h_c \tag{11}$$

$$z_{0c} = 0.264 \times (h_c - d_c) \tag{12}$$



The resistance to heat transfer is take as resistance to momentum under neutral conditions. We neglected corrections for stable and unstable conditions and extended use of Eqs. 11-12 to non-neutral conditions.

– the canopy stomatal conductance submodel is based on a hypothetical maximum conductance, $g_{s,max}$, which is modified by empirical stomatal response functions which vary between zero and unity. These functions are combined in a multiplicative polynomial equation (Jarvis, 1976) to model the responses to the air $CO_2$ concentration, air vapour pressure saturation deficit, $\delta e_w$ , the incident shortwave radiation, $SW$, and the leaf water potential, $\psi_{leaf}$ . Since the leaf water potential depends on the tree hydraulic conductivity (Eq. 21), this model accounts for the effects of plant height on stomatal conductance (Delzon et al. 2004). The stomatal response modeled is therefore independent of the photosynthesis rate and allows for putative nighttime positive values. The stomatal model includes a time constant which accounts for the response time of stomata to changing climatic or leaf water potential conditions. The individual stomatal response functions used are allowed to vary according to the species considered.The steady state stomatal conductance, $g_{s,c}*$ and its dynamic counterpart, $g_{s,c}$ are :

$$g_{s,c,h} * = g_{s,max} \times f_{SW} \times f_{\delta e} \times f_{\psi} \times f_{CO2}$$

$$g_{s,c,h}(= \frac{1}{r_{s,c,h}}) = g_{s,c,h} * + (g_{s,c,h-1} - g_{s,c,h}*) \times \exp{(\frac{-1}{\tau})} \tag{13}$$

The sensible heat flux from canopy layers and the soil is

$$H_c = \frac{1}{r_{Hc}} \times \rho_a \times c_p \times (Ts_c - Ta_z) \tag{14}$$

Since mass transport from the canopy layers and soil into the atmosphere are essentially turbulent, resistance and conductance for heat, momentum and mass transport will not be distinguished further in this section. The wet and dry fractions of each canopy and soil layers are calculated dynamically using the Gash's canopy water balance model resolved at an hourly time step (Eqs. S11-S14, Gash (1979)). This model, which needs few parameters, estimates the interception of incident rainfall by the canopy and the depth of water retained on the canopy. The tree trunks are assumed to behave as the canopy layer during leafy periods when the canopy is in leaf (Table S1). Under wet canopy conditions the stomatal resistance is assumed to be zero and, the flux of water vapour exchanged between the canopy layers and atmosphere is transferred only across the aerodynamic resistance : :

$$\lambda \times E_{wet,c} = min(S_{W,c}, \frac{(1 - f_{dry,c}) \times \rho \times c_p \times [\delta e_w + s \times (Ts_c - Ta_z)]}{\gamma \times r_{a,c}}) \tag{15}$$

where $S_{W,c}$ is the water stored at the surface of the canopy and $(1 - f_{dry,c})$ is the fraction of canopy that is wet. In the case of condensation, i.e. when $\lambda E_{wet,c} < 0$, the corresponding amount is added to the rainfall and directed to the lower canopy layer. The canopy temperature is not differentiated between the wet and dry fractions.

The canopy stomatal resistance is added to estimate the vapour flux emitted from the dry canopy i.e. the plant transpiration:

$$\lambda \times E_{dry,c} = \frac{f_{dry,c} \times \rho \times c_p \times [\delta e_w + s \times (Ts_c - Ta_z)]}{\gamma \times (r_{s,c} + r_{a,c})} \tag{16}$$





The soil is treated using a specific surface resistance calculations as follows:

$$r_{s,soil} = 100 \times \frac{\theta_{SAT} - \theta_{WP}}{\theta_A - \theta_{WP}} - 1 \tag{17}$$

The resulting latent heat flux, $\lambda E$, is the sum of dry and wet evaporation over the two canopy layers and soil.

## 2.5 Water transfer

The soil is partitioned into three horizontal layers which are defined by their respective water content and may therefore have a variable depth and thickness:

- the top layer $A$ is unsaturated, i.e. , its water content $\theta_A$, varies between the wilting point, $\theta_{WP}$ and maximal water holding capacity, i.e., the field capacity, $\theta_{FC}$;

- the water content of the layer $B$, $\theta_B$, is between the field capacity and saturation, $\theta_{FC} \leq \theta_B < \theta_{SAT}$ , and $z_{AB}$ is the
lower level of layer $A$ (upper level of layer $B$);

- the layer $C$ is saturated at $\theta_{SAT}$ and $z_{BC}$ is the lower level of layer B (upper level of layer $C$).

Water is transferred from the soil surface into the three layers according to a 1-D cascading formalism through either (i) as frontal diffusion or (ii) fast gravitational transfer and according to a simple bucket model. Because the water content of $B$ and $C$ cannot vary - only their thicknesses can vary - the layer $A$, if present, is first filled up until field capacity, further water input
is then transferred to the layer $C$ that is filled until it reaches the soil surface when $z_{BC} = 0$. In the absence of sufficient plant water uptake, deep runoff of groundwater occurrs; this depends on the local topography and hydrological environment and is modelled as :

$$D = D_{max} \times \left( \frac{z_{min} - z_{BC}}{z_{min}} \right)^{k_w} \tag{18}$$

where $D_{max}$ is the maximal drainage rate which will occurr when the water table is at the soil surface, $z_{min}$ is the depth at
which drainage of the water table ceases and $k_w$ a shape parameter describing the attenuation of drainage rate with the water table depth. In this equation, the depth is counted as a positive number.

The soil evaporation is emitted from the upper layer, that is either $A, B$ or $C$. Plant transpiration is taken from the soil layers above the maximal root depth according to their respective water availability, first from the saturated layer $C$ , then and if necessary from the intermediate layer $B$ and finally from the upper layer $A$. Hence, when soil is saturated, i.e. $z_{BC} = 0$ and
layers $A$ and $B$ do not exist, the transpiration uptake lowers the level of $C$ and creates a layer $B$ until $z_{BC}$ passes beneath the root level i.e. $z_{BC} < z_{roots}$. The transpiration is then taken from the layer $B$ until its water content, $\theta_B$, drops down to the field capacity, $\theta_B = \theta_{FC}$. Layer $A$ is then created and transpiration is taken from $A$.

The water withdrawn by plants is transferred from the soil to the roots and from the roots up to the canopy layers along a series





of two hydraulic resistances, the soil-to-root resistance, $r_{soil}$ and the mean root-to-foliage resistance, $r_{xyl}$.

$$r_{soil} = \frac{[1 + (\alpha_{VG} \times \psi_{soil})^{n_{VG}}]^{m_{VG}/2}}{\{1 - (\alpha_{VG} \times h_P)^{n_{VG}-1} \times [1 + (\alpha_{VG} \times \psi_{soil})^{n_{VG}}]^{-m_{VG}}\}^2} \tag{19}$$

$$r_{xyl,c} = k_{x0} + k_{x1} \times h_c^{k_{x2}} \tag{20}$$

A plant bulk capacitance, $C_T$, is added in derivation of the two-resistance pathway (Eq. S15 from Loustau et al. 1998). Having defined a global soil-to-foliage resistance, $r_c$, as $r_{soil} + r_{xyl,c}$, the canopy foliage water potential is:

$$\psi_{c,t} = (\psi_{soil,t-1} - \frac{E_{dry,c} \times r_c}{LAI_c \times 3600}) \times [1 - \exp{(-\frac{\delta t}{r_c \times C_T})}] + \psi_{c,t-1} \times \exp{(-\frac{\delta t}{r_c \times C_T})} \tag{21}$$

### 2.6 Carbon cycle

The carbon cycle includes a suite of processes starting with the $CO_2$ uptake from the atmosphere by photosynthesis in the
canopy layers and continuing with the subsequent transport and metabolic processes until carbon is exported out of the ecosystem, being either returned into the atmosphere by the respiration of the vegetation or soil, leached as dissolved carbon in groundwater, or exported during harvest (Fig. 1). Methane fluxes, the emission of volatile organic compounds and herbivory are neglected in version 3.0 of the model.

#### 2.6.1 Photosynthesis

The photosynthetic carbon uptake by each canopy layer is formalised in GO+ following Farquhar et al. (1980) and de Pury and Farquhar (1997) as the minimum of the RubP (Ribulose -biPhosphate) regeneration by electron transport and its carboxylation rate by RubisCO. The effects of leaf nitrogen and phosphorus content on photosynthesis are not implemented in the version 3.0 of the GO+ model and so are not presented here . The carbon assimilation is calculated separately for shaded and sunlit fraction of the foliage, denoted by subscript $s$, following the same set of equations (Eqs. 23, S17-S20).
The temperature dependency of the maximal rates of carboxylation by RubisCO and electron transport, $V_{cmax,c}$ and $J_{max,c}$ are computed according to Medlyn et al. (2002) (Eqs S22-S28). The chloroplastic concentration in $CO_2$, $c_x$, is estimated from the atmospheric concentration $CO2_a$, accounting for a series of three resistances from atmosphere to chloroplast, the aerodynamic resistance (Eq. 3), stomatal resistance (Eq. 13) and leaf internal resistance, the latter being taken from Ellsworth et al. (2015) (Eq. S21). The combination of the $CO_2$ transport equation $Anet_{c,s} = g_{CO2,c} \times (CO_{2a} - c_{x,c,s})$, where the total conductance
to $CO_2$ is $g_{CO2,c,s} = \frac{1}{r_{H,c} + r_{s,c} + r_{m,c}} \times \frac{D_{CO2}}{D_{H2O}}$, with biochemical reaction rates Eqs. (S18-20) leads to a quadratic equation in which has the solution:

$$Anet_{c,s} = \frac{b - \sqrt{b^2 + 4 \times c}}{2} \tag{22}$$

with

$$b = \begin{cases} g_{CO2,c,s} \times (CO_{2a} + K_m) + V_{cmax,c} - R_d & \text{if } W_c > W_j \\ g_{CO2,c,s} \times (CO_{2a} + 2 \times \Gamma^*) + \dfrac{J_{c,s}}{4} & \text{otherwise.} \end{cases}$$





and

$$c = \begin{cases} g_{CO2,c,s} \times [(CO_{2a} + K_m) \times R_d - (CO_{2a} - \Gamma^*) \times V_{cmax,c}] & \text{if } W_c > W_j \\ g_{CO2,c,s} \times [(CO_{2a} + 2 \times \Gamma^*) \times R_d - (CO_{2a} - \Gamma^*) \times \dfrac{J_{c,s}}{4}] & \text{otherwise.} \end{cases}$$

where the electron transport rate $J_{c,s}$, is calculated according to Eq. S19. The net photosynthesis is then integrated at canopy layer level using the shaded and sunlit area fractions of foliage $LAI_{sun}$ and $LAI_{shade}$ (Eq. S1) and foliage temperature for estimating $K_m$, $V_{cmax,c}$, $J_{max,c}$ and $\Gamma^*$ (Eqs S22-S28). At the ecosystem level, the net assimilation of $CO_2$ and the gross primary production by the canopy foliage are therefore, respectively:

$$A_{ECO} = \sum_{c=1,s=0}^{2,1} Anet_{c,s} \times LAI_{c,s}$$

$$GPP_{ECO} = A_{ECO} + \sum_{c=1}^{2} R_{d,c}$$

### 2.6.2 Respiration

The respiration from living plants, $R_a$, is assessed as a mass flux of $CO_2$ released into the atmosphere. It is partitioned between a growth component and maintenance component. The growth respiration $R_g$ is estimated as a fixed fraction of the carbon allocated to growth that depends on the chemical composition of the organ, leaves, branches, stems, roots (Penning de Vries et al. , 1974). The maintenance respiration, $R_m$, is a basal metabolic rate of respiration that depends on the living biomass and temperature. It is calculated separately for above-ground parts and below-ground parts as follows.

– The foliage respiration of each layer, $Rm_{c,f}$ is :

$$Rm_{c,f} = LAI_c \times R_{d,T15,c} \times \exp\left(E_a(R_d) \times k_{T,c}\right) \tag{23}$$

where $k_{Tc}$ is a temperature factor also used for the parameters representing the temperature dependency of photosynthesis (Eq. S22).

– The maintenance respiration of other tree parts ( stem, branches, taproot, coarse, small and fine roots) is calculated on the basis of the mass of nitrogen in living biomass , $N_c^*$ (Dufrêne et al. 2005).

$$Rm_{T15,c} = N_c^* \times R_{N,T15} \tag{24}$$

where $R_{N,T15}$ is the rate of maintenance respiration per unit mass of nitrogen (Ryan, 1991). The calculation of $N_c^*$ is resolved at the tree level as detailed in the supplementary material Eqs. S29-33. The temperature dependent respiration integrated over the entire layer, $Rm_{w,c}$, is then:

$$Rm_{w,c} = \sum_c Rm_{T15,c} \times Q_{10}^{\dfrac{T_{air} - 15}{10}} \tag{25}$$

where $Q_{10}$ is multiplier of maintenance respiration for a 10°C temperature increase.





– The maintenance respiration of the understorey components, foliage, roots and perennial part, is depending on their only biomass and is using the same temperature response than trees.

### 2.6.3 Carbon allocation and growth

The GO+ allocation scheme allows a flexible allocation of carbon among individual trees and between above-ground and below-ground tree parts. The allocation scheme of the understorey is fixed. The allocation scheme is summarised in Fig. 2. The carbon allocation between below-ground and above-ground parts is regulated by a water stress index. Subsequently, the carbon is distributed among plant parts based upon empirical allometric equations. .

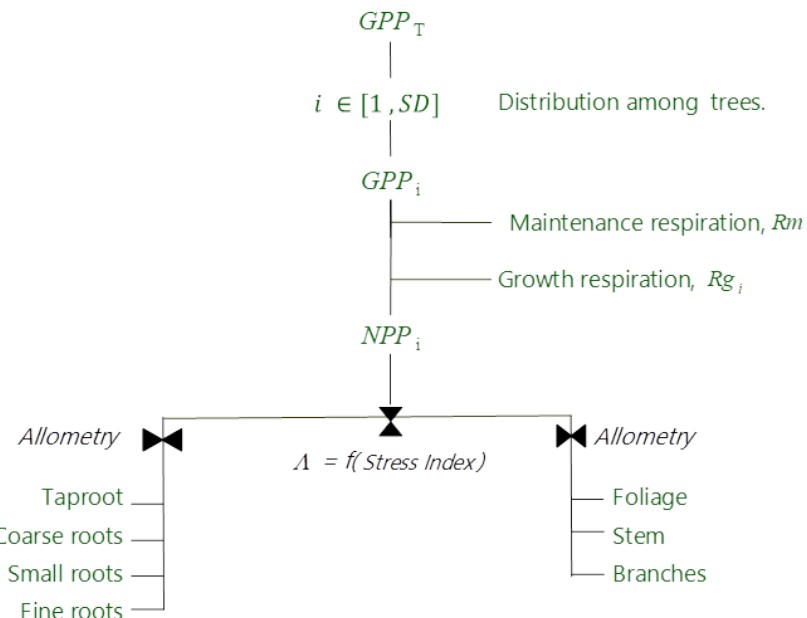

**Figure 2.** Allocation scheme of carbon for the tree canopy in the GO+ v. 3.0 . $SD$ is the number of individual trees per ha and $NPP_i$ the net primary production allocated to an individual tree $i$.

– For the tree stand, the growth is resolved at a daily time-step for the foliage and at an annual step for the stems, branches, taproot, coarse roots and small and fine roots. The individual tree growth is modelled following a three-step process.

– (1) The carbon uptake by photosynthesis $GPP_T$ is shared among individual trees according to their respective contribution, $\lambda_i$, to the canopy foliage dry mass $W_{L,c}$.

$$\lambda_i = \frac{W_{L,i}}{W_{L,c}} \qquad (26)$$



where $i \in [1, SD]$ and $\sum_{i=1}^{SD} \lambda_i = 1$, $SD$ being the number of stems per unit area. For each individual the amount of carbon allocated to growth, $NPP_i$, is the gross primary production after the respiration of foliage, woody parts and roots have been subtracted, $NPP_i = \lambda_i \times GPP_T - [Rm_{a,i,f} + Rm_{a,i,w,} + Rm_{r,i} + Rg_i]$.

– (2) $NPP_i$ is partitioned between above-ground and below-ground parts using an root / shoot allocation coefficient $\Lambda$. This coefficient depends on the annual water stress index, $I_{\text{stress}}$, that is related to the ratio of the annual tree transpiration, $E_{\text{dry, T}}$, to the potential transpiration $E_{\text{pot, T}}$. The potential transpiration, $E_{pot,T}$, is calculated with a stomatal model having only $SW$ and $CO_2$ limitations and corresponds to the transpiration of a canopy unlimited by hydrological or meteorological drought.

$$\Lambda = k_{\lambda 1} \times \exp(k_{\lambda 2} \times I_{\text{stress}}{}^{k_{\lambda 3}}) \tag{27}$$

with

$$I_{stress} = 1 - \frac{E_{dry,T}}{E_{pot,T}} \tag{28}$$

The allocation scheme allows therefore a shift of the annual amount of carbon allocated to growth of below-ground parts, $dW_{r,T}$, when the stress index increases (Landsberg and Waring, 1997). The annual net amount of carbon available for the structural growth of roots is calculated as:

$$dW_{r,i} = \Lambda \times NPP_i \tag{29}$$

The corresponding amount of carbon allocated to above-ground structural parts is:

$$dW_{a,i} = NPP_i - dW_{r,i} \tag{30}$$

The individual tree biomass above-ground and below-ground $W_{a,i}$ and $W_{b,i}$ are then updated :

$$W_{a,i,year+1} = W_{a,i,year} + dW_{a,T,year}$$
$$W_{b,i,year+1} = W_{b,i,year} + dW_{b,T,year} \tag{31}$$

– (3), GO+ allocates the amount of carbon available for above-ground growth among foliage, branches and stem and for below-ground parts among taproot (stump + main pivotal root), coarse roots (diameter > 20 mm) , small roots ( diameter between 2 and 20 mm) and fine roots (diameter < 2 mm) using species-specific sets of allometric equations. Each set of values is specific to the tree species considered. Such equations link the stem diameter at breast height, $D_{130}$, to the biomass of aerial parts. The $D_{130}$ is substituted from the set of allometric equations so that each compartment biomass can be related to the total above-ground biomass. The foliage growth is distributed over the next growing period meaning that the current cohort of leaves relies upon the previous year $NPP_T$. This implies that the current year $LAI$ depends on the previous year $NPP_T$ and stress index. The growth of the other parts of each tree is not dynamic but is calculated at a yearly resolution; it is instantaneously updated at the end of the year. The equations used for maritime pine, European beech and Douglas fir are shown as examples in Eqs. S34-S62. The height of individual trees is also derived from allometric equations.




— The understorey allocation scheme is resolved dynamically at a daily time step using two ordinary differential equations. We assume the horizontal distribution of the understorey vegetation is uniform and no individual plants are defined. The vegetation includes three compartments, the foliage, $f$, roots, $r$, and perennial parts, $p$. The understorey growth comprises two processes, growth and mortality, that are applied to each compartment, foliage, roots and perennial parts with specific parameter values. The growth of understorey biomass parts is resolved at a daily time step as the minimum of a demand and a supply functions, $dW_{d,j}$ and $dW_{s,j}$ respectively.

— The demand function of each compartment, foliage, roots and perennial parts, $dW_{d,j}$ at day $DOY$ is the derivative of the sigmoid function $s_j$ times the asymptotic value of biomass $W_{max,j}$:

$$dW_{d,j} = W_{max,j} \times s_j \times (1 - s_j) \tag{32}$$

$$s_j = \frac{1}{1 + exp[-k_p \times (DOY - DOY_{0.5,j})]}$$

where $dW_{d,j}$ is the daily potential biomass increment of compartment $j$, $k_p = \frac{1}{GD} \times 2 \times Log(\frac{1}{k_s} - 1)$ where $GD$ is the maximal growth duration, $k_s$ a flattening coefficient (kurtosis) and $DOY_{0.5,j} = BB_j + \frac{GD}{2}$ the day of year by which half of the growth has been achieved, $BB_j$ being the day when growth starts.

— The supply function of compartment $j$, $W_{s,j}$, is the pool of carbon available for growth. Its is fed by the fraction of the carbon allocated to the compartment, $dW_{s,j}$ calculated as:

$$dW_{s,j} = \lambda_j \times (GPP_U - R_{m,U}) \times \frac{1}{1 + Rg_j} \tag{33}$$

where $\lambda_j$ is an allocation coefficient to compartment $j$ and $Rg_j$ the respiration cost associated with the compartment $j$. The $NPP_U$ allocation among the three compartments is fixed by three parameters, $\lambda_j$, subscript $j$ standing for $f$, $p$ or $r$. The growth starts at the "budburst day", $BB_j$, according to a simple model of accumulated "degree-days" and is paused when the soil moisture deficit or air temperature drop below a fixed threshold value of $SMD_{GU}$ or $T_{GU}$ respectively.

## 2.7 Vegetation phenology

**Leaf unfolding, senescence and growth**

A specific phenological model of leaf development can be specified for any tree species comprising the overstorey layer. This is illustrated for three phenological model types in the supplementary material (Table S3). They include (i) a simple thermal time model (maritime pine), (ii) a parallel model combining simultaneously chilling and forcing temperatures (Douglas fir) and (iii) an alternating model assuming a negative exponential relationship between the sum of forcing units required for completing





the quiescent phase and the sum of chilling units received (European beech). A single model is implemented to describe the phenology of the understorey vegetation. It includes a simple thermal time model for leaf unfolding with parameters that are identical for the three compartments, foliage, perennial part and roots (Table S5). The temperature used for accumulating degree-days is the air temperature for above-ground parts and the soil temperature at 0.1 m depth for the roots.

**Senescence**

The senescence of the different tree and understorey parts is modelled according to the organ-specific turn-over time and to the mortality induced by low temperature, soil moisture deficit or date respectively(Tables S4-S5). The timing of senescence is fixed for the cohort of coniferous needles. For broadleaf species it is a linear function of the sum of the mean daily shortwave radiation( $= \frac{1}{24} \sum_0^{24} SW \downarrow$, in W m$^{-2}$ ) accumulated from the date of budburst until DOY=258 for European beech, Table S4),

as fitted on data provided by the French ICP forest network (http://icp-forests.net/) from 14 beech stands where meteorological data were recorded (Le Bourgeois 2008). This accumulated radiation model explains 60% of the variance of the leaf senescence date across the data set explored; it compares well with other modelling attempts requiring more parameters and variables (Delpierre et al. 2009). The understorey senescence is triggered by low temperature, soil moisture deficit or date: beyond a fixed threshold, the understorey mortality is set at a fixed rate (Tables S1, S5). The separation of dead parts from the mother

plant occurs as a single event either annually at the end of year for tree branches and roots or daily for the tree and understorey foliage. After separation, dead parts are immediately incorporated into the soil.

**Mortality**

Apart from the management operations (spacing, thinning, clear-cutting), the process of mortality of individual trees is diverse, complex and poorly understood and documented: it is therefore not modelled in version 3.0 of GO+. Instead, at the

end of each year, the carbon balance of individual trees is computed as the difference between individual $NPP_i$ and the senescence of branches and roots. A "natural" tree death occurs when the carbon balance of a tree is negative, i.e., the net amount of carbon allocated for growth is negative. This is mainly provoked by combinations of strong soil water deficit, air water vapour deficit and high temperatures. The understorey cannot "die" naturally but is maintained as a perennial carbon pool that can be regarded as a survival form (seeds, rhizomes, bulbs etc.). This allows regrowth of ground vegetation after

clear-cutting. Following natural mortality, thinning or clear-cutting, the parts of harvested trees and understorey that are not exported are added to the soil pool. In particular, the part of the ground vegetation composing the understorey that is destroyed by forest operations such as soil preparation and possible disking prior to tree spacing or thinning interventions is added to the soil.

## 2.8 Soil carbon

The Roth-C v 6.3 model is implemented in GO+ with only a few modifications (Coleman and Jenkinson 1996). Only one soil layer is considered for soil carbon and the entire organic carbon stock of the soil is assumed to be included between the soil surface and the soil depth down a vertical profile modelled as exponentially decreasing with depth (Arrouays and Pelissier,1994). The inputs of organic matter to the soil are incorporated at the time of death — or harvest — when plants die, or at the time of separation from the mother plant for the senescing parts of foliage, branches, stems and roots. Mineralization





and decomposition processes are discretized at an hourly time step and forced by the soil temperature at average depth where the respiration occurs,$T_{S,Rh}$ and soil moisture in layer $A$. The temperature at the average soil depth where the heterotrophic respiration occurs, $T_{S,Rh}$ is estimated using an empirical force-restore model depending on air and soil reference temperature as follows:

$$T_{S,Rh} = T_{S,Rh} + k_{Ta} \times (T_a - T_{S,Rh}) + k_{Tref} \times (T_{ref} - T_{S,Rh}) \tag{34}$$

The main adaptation introduced concerns the impact of forest operations on mineralization and decomposition rates as described in the next section.

### 2.9 Management: forest operations and harvesting, nutrient balances, wood products.

The management module of GO+ is separated from the core biophysical and biogeochemical modules. Management intervenes during the model execution as a suite of operations affecting processes involved in the soil carbon dynamics or affecting the understorey layer and tree stand. The forest management schemes are described as itineraries starting from regeneration and running until the next clear-cut thus covering the entire life cycle of the tree stand. Two main management strategies are implemented in the GO+ 3.0 version, coppicing and regular stand. So far the former has been used only for eucalyptus whereas regular stand management is the main strategy used for pine and fir species. This versatility allows the GO+ model to describe the main management schemes used in monospecific even-aged forests, from short rotation eucalyptus coppice to stands of coniferous or broadleaved species, unmanaged old-growth forests (self-thinning) and agroforestry systems (coffee plantations). The model results can therefore be used for analysing the interactive effects of management and climate change on forest energy, water and carbon balances as well as commercial production. Further developments that will account for tree species mixture and irregular forests are ongoing but not yet implemented in version 3.0.

### 2.9.1 Soil preparation

Although Roth-C was initially calibrated for arable soils subject to periodic ploughing, it may underestimate the abrupt effect of ploughing on forest soils (Balesdent et al. 1998, Gottschalk et al., 2010). In managed forests, soil preparation may include techniques such as tillage, moulding and disking which may occur at only decade-long time intervals and therefore induces some drastic changes in the structure and microclimate of the upper soil horizons and organic layers. This may explain the effects of the preparation of forest soils on mineralization (Wang et al. 2018) and decomposition of the soil organic matter (Chen et al. 2004). In the GO+ model, we introduced a ploughing effect specifically for forest soils.. With this scheme the effects on the soil carbon of the preparation techniques such as ploughing, moulding and disking can be prescribed in the management module at any specific time during the rotation, e.g., after clear-cut, before every or specified spacing, thinning and clear-cutting operation or before regeneration. Immediately after any operation affecting the soil, the mineralization and decomposition rates of the soil carbon fraction affected are enhanced, this enhancement then decreases exponentially with time. Fig.3 shows the dataset taken from Jolivet (2000) which is used for calibrating the enhancement factor and its life half-life. The Table S1 provides the default values of the parameters. This approach is simple but easier to implement on multiple sites and





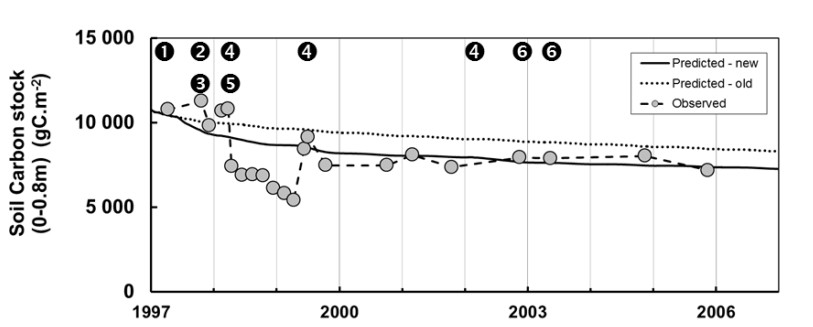

**Figure 3.** Changes in the soil organic carbon stock during the regeneration phase following a clear-cut of a maritime pine stand as simulated by the GO+ model with and without adaptation for soil preparation (full and dotted lines respectively) and measured in the field (grey dots). Data taken from Jolivet (2000). The numbers inset in black dots refer to the forest operation. 1: Clear-cutting and logging; 2: Heavy disking; 3: Stump removal; 4: Cover crop; 5: Tillage; 6: Vegetation crushing.

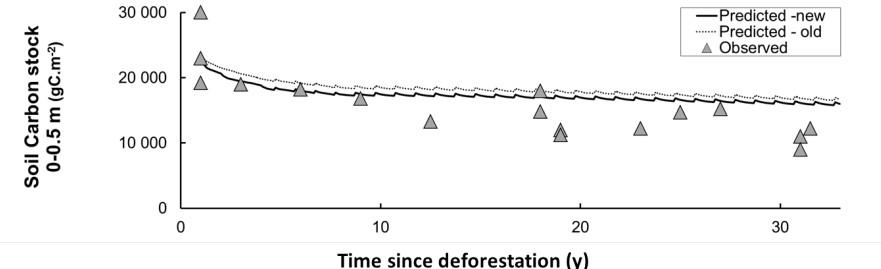

**Figure 4.** Changes in the soil organic carbon stock after deforestation of a maritime pine stand (at year=1) and continuous cropping of maize from year=3 onwards. Data points show carbon stocks from 0 to 0.5 m depth collected at different sites in southwest France presented as a chronosequence since deforestation (Arrouays and Pelissier, 1994).

spatial scales than the more mechanistic Gottschalk et al. (2010)'s which differentiates the ploughing effect according to the carbon pools described in Roth-C and to their linkage with the mineral fraction. We also evaluated the model on soil carbon data collected by Arrouays and Pelissier (1994). Those data provide a time series of soil carbon stocks following deforestation and continuous maize cropping in Les Landes forest in southwest France. The difference between the original version of Roth-C and the GO+ version is substantial, i.e., 5 to 12% of the total modelled soil carbon; this difference is maintained over time. The simulations output from the improved GO+ version are closer to the observations for both the short-term changes observed during soil preparation (stump removal, slash burial, vegetation crushing) (Fig.3) and long term soil carbon chronosequence following deforestation (Fig. 4).





### 2.9.2 Tree stand management

The tree stand management has a dramatic impact on forest ecosystems and their functioning. The model GO+ describes mechanistically the effects of the main management alternatives applied to even-aged monospecific forest stands that dominate European forests. To this end a large framework of forest operations is implemented in the model and can be assembled to construct different technical itineraries. The operations prescribed in a given itinerary are triggered according to forest management rules as follows.

– The stand regeneration can result from either natural processes, sowing or planting, the number of seedlings and their age and size distribution being flexible.

– The tree harvests are defined by the number and size of trees felled at each thinning and the final clear-cut. Successive spacings, coppicing, thinning and final clear-cutting occur either at given stand ages; they can be triggered by a competition index (Le Moguedec and Dhote 1992, Bellassen et al. 2010, 2011, Guillemot et al. 2014), or by target values of stand variables commonly used in forestry such as the mean tree diameter and height, stand basal area, or mean diameter and height of the 100 biggest trees per hectare at a given age. The selection of trees to be felled is flexible and can be either random, from the top, i.e., the bigger trees, or from below. A wide range of thinning strategies of varying complexity can thus be simulated by the model from the relative density index used for broadleaved species to the application of the "natural" self-thinning rule (Reineke, 1933).

– Specifying which tree parts are to be harvested may be any combination of stemwood, branches, foliage, stumps and roots. The harvest residues are input into the soil. GO+ predicts the size distribution of the stems harvested thus allowing raw wood products to be routed into life-cycle models, such as the C.A.T. model (Pichancourt et al. 2014), at large spatial scales.

– Coppicing is modelled as a clear-cut followed by the resprouting of a variable number of stems, which grow from the stumps left behind. The growth of the new stems is fed by a carbon pool that corresponds to the basal part of the stem having a diameter of $1.2 \times D_{130}$ and a variable height (default value is 0.1 m) that is assumed to be residue. At this stage, the allocation of $NPP$ to the above-ground part is increased until the root/shoot ratio is restored to its equilibrium value ($k_{\lambda 1}$, Eq. 29). This allows the stand $LAI$ to increase rapidly after cutting, as is observed for coppices.

The Fig. 5 illustrates the impacts on the biomass and soil carbon stocks of typical management cycles implemented in GO+ and applied commonly in European forestry. The coniferous and broadleaf standards are managed according to "Long", "Short" and "Standard" rotations. The eucalyptus coppice includes one ("Long") or two ("Short" and "Standard") cuttings between each plantation, the "Short" option having a smaller diameter threshold for cutting than the "Standard" option. The levelling off of the beech biomass with stand age in the long, and to a lesser extent in the standard, options is mainly provoked by a decline in $NPP$ due to increased biomass respiration but also by a decrease in $GPP$. The predicted levelling-off of production is less marked or absent for other species and management options because the thinning regime prevents the tree stand biomass



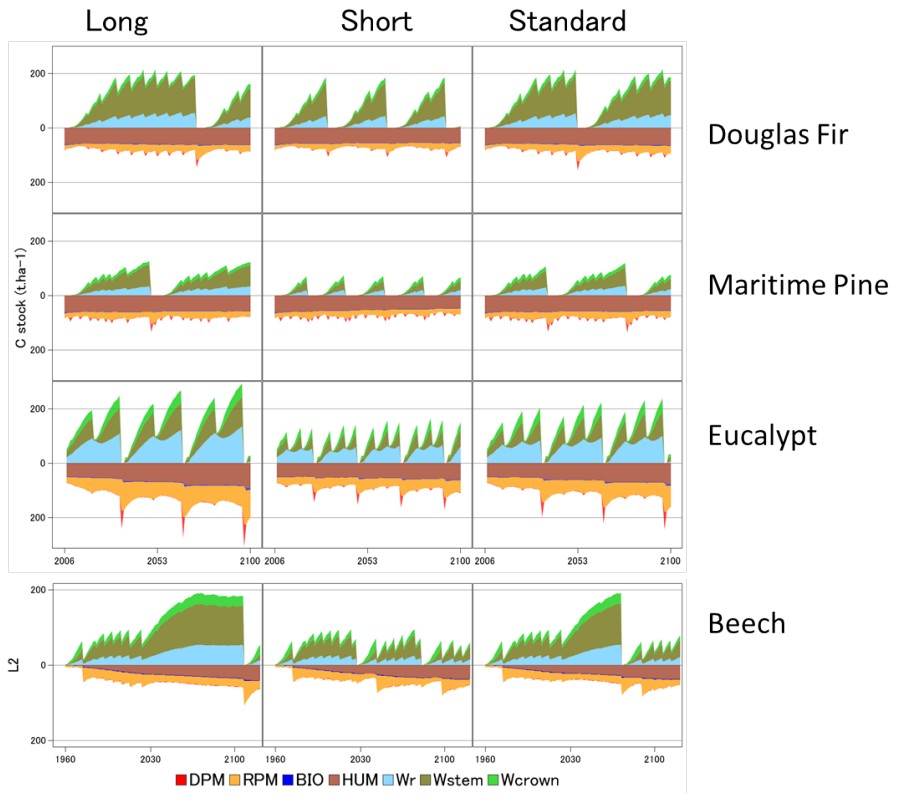

**Figure 5.** Biomass and soil carbon stocks simulated for four species and three management alternatives. The simulations were forced by the RCP 2.6 climate scenario. The grid point location is close to the centre of the French geographical distribution of each species. The pine and Douglas fir are grown in plantations managed with thinning rules and a final clear-cut based upon the mean stem diameter. Harvested parts are the stem only (Long) or crown and stem (Short and Standard). The eucalyptus is managed as coppice with two cuttings of sprouts before new planting. The stump age is used to trigger coppicing and final cut. The beech stand is managed according to the relative density index (Le Moguedec et Dhôte 1992). In the examples shown, the beech stand simulated was regenerated on a bare soil with low organic matter content and no understorey. The soil fractions "BIO" and "DPM" have low values that are barely visible.

to saturate. Apart from the beech stand that was simulated on a bare soil, the soil carbon dynamics is mainly marked by the periodic massive input of resistant plant material leftover following harvest operations. The soil carbon dynamics contrast sharply with the forest management options for the eucalyptus coppice and much less for the other species.

An application of the model at the country level is illustrated by Fig. 6 where two afforestation scenarios, the "Short" and "Standard" alternatives, were run from 2006 to 2100 in dynamic mode under RCP 4.5, starting from cultivated soils with low organic content. The short rotation is cut at 25 years and includes deep ploughing and fertilisation, the Standard rotation is cut at 50 years and includes partial tillage. The simulation covers the whole French metropolitan area at 8x8 km resolution (9600 pixels) and is shown only as an illustration, all simulated pixels being afforested simultaneously.





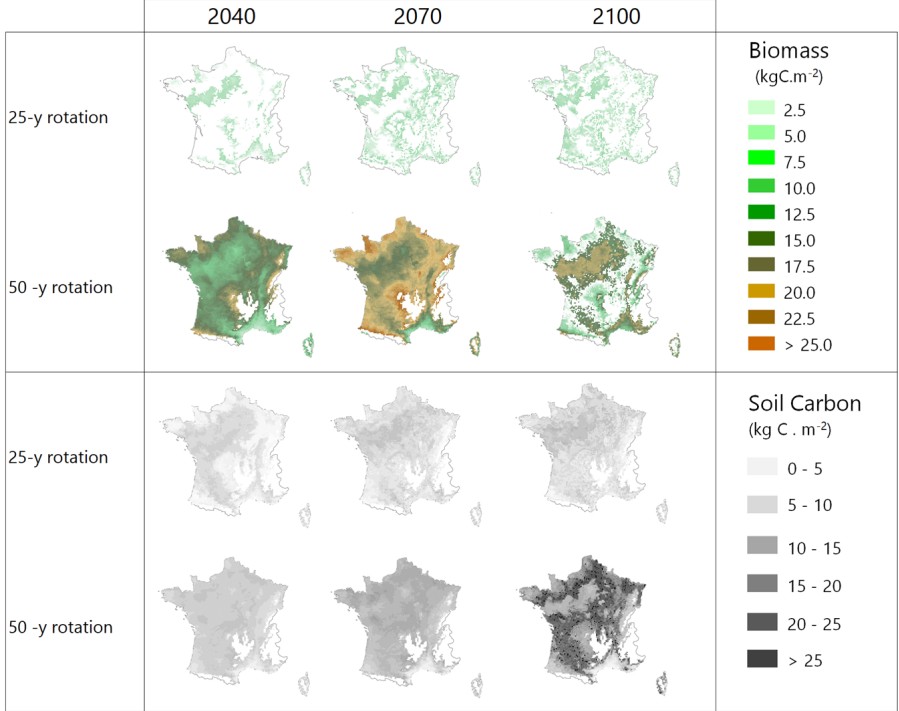

**Figure 6.** Biomass and soil carbon stocks of maritime pine stands simulated over the entire French metropolitan area for two management alternatives under climate scenario RCP 4.5. GO+ was run dynamically from 2006 to 2100 and initialised on bare soils with new stands in 2006, mimicking as the afforestation of cultivated soils.

### 2.9.3 Vegetation control

The vegetation management operations are described in terms of area affected and fraction of the understorey vegetation biomass destroyed. For releasing the trees from vegetation competition for light, water and nutrients, or during soil preparation a variable fraction of ground vegetation is affected and the corresponding fractions of the above-ground and below-ground understorey biomass are assumed to be destroyed and added to the soil carbon pool (Subedi et al. 2014). Prior to spacing, thinning or clear-cutting, a variable fraction of understorey biomass is also prescribed to be destroyed. For instance, in the pine forests of southwest Europe, rolling heavy disk trails is a common practice at plantation and before each thinning or clear-cutting. These disking operations are applied between rows of trees on three quarters of the soil surface area and affect typically 15% of the soil carbon. The model simulates this practice through the following:

- mortality of 75% of the above-ground biomass (foliage and perennial parts) and 50% of the below-ground biomass (roots) of understorey vegetation;

- as described previously in section 2.9.1, a three-fold increase in the mineralisation, decomposition and conversion-into-$CO_2$ parameters of the Roth-C model for 15% of the soil carbon with a half-life of 92 days.





### 2.9.4    Nutrient export

Achat et al. (2018) provide a detailed description of the nutrient module that was recently added to the core GO+ model in order to quantify the export of nutrients from the ecosystem through harvesting and soil preparation. This module evaluates the nutrient (N, P, K, Ca and Mg) stocks in standing tree biomass and soil. The nutrient outputs from these stocks through

biomass harvesting can then be calculated. In short, this module calculates the main nutrient content of the soil, tree and understorey parts from literature values and Go+ predicted values of biomass and soil components. This calculation is based on allometric equations which account for the age and size of each tree part allowing the nutrient content of trees to vary with age and size. Realistic estimates of the nutrient exports related to forest practices can thus be produced under a range of climate-management combinations, as is illustrated by Achat et al. (2018). In their simulation, the harvested tree parts were

allocated to size categories, allowing them to predict the nutrient balance of management schemes according to the harvest intensity.

### 3    Verification and parameterisation

### 3.1    Verification

The verification tests consisted of checking the conservation of energy and mass of carbon and water for a long time series

of model simulation. The period covered a typical forest stand rotation from the seedling stage to the final clear-cut; thinning and the impact of extreme natural events were included. We selected the Le Bray site to provide the benchmark data for the sensitivity analysis and evaluation of the model. The tree stand demography at this site was monitored from 1987 to 2008, with measurements of sensible heat, $CO_2$ and $H_2O$ fluxes and meteorological variables starting in 1996. The period starts in 1984 and ends in 2010. It includes a series of dry years (1989–1991, 2002–2003, 2005–2007) and the December 1999 "Klaus"

hurricane that fell or broke 22% of the trees. The model was run from 1984 to 2001 forced with meteorological data measured at the French synoptic network station being interpolated across the 8x8 km SAFRAN grid. The number and size of the trees thinned and felled for this period in 1991, 1996, 2001 and 2005 were also used to prescribe the thinned and windthrown trees. The verification test results are summarised in Table 2.

–   The average hourly gap in the energy balance $R_n = H + LE + G$ was $8\ Wm^{-2}$, that is 9%. This gap results from the

extension of neutral regime to stable and unstable conditions which results primarily in a slight underestimation of the convective heat fluxes, $LE$ and $H$. The Nakai's model for estimating roughness length and displacement height leads to underestimate $H$ for low value of Leaf Area Index that is below $1.5\ m^{-2}\ m^{-2}$.

–   For the water balance, we checked independently that the annual amount of precipitations from 1984 to 2010, $Rain$, was correctly allocated among (i) interception by the canopy and soil layers, $E_{wet}$, vegetation transpiration, $E_{dry}$, ground-

water discharge or runoff, $D$, and the variation of the soil water stock over this period, $\Delta(\theta_{rootlayer} \times z_{root})$. The discrepancy found was $5\ mm\ yr^{-1}$ over a total amount of $960\ mm\ yr^{-1}$ that is 0.5%.





**Table 2.** Verification tests operated on the model. The test is a simple conservation test applied to the annual values of energy, water and carbon fluxes over the 1984 to 2010 period.

| Test | Input | Output | | | | | | Balance (I-O) |
|---|---|---|---|---|---|---|---|---|
| Energy balance | $Rn$ | $H$ | $\lambda E$ | $G$ | | | | |
| (W m$^{-2}$) | 89 | 52 | 29 | 0.0 | | | | 8 |
| Water balance | $Rain$ | $E_{wet}$ | $E_{dry}$ | $\Delta(\theta_{rootlayer} \times z_{root})$ | $D$ | | | |
| (kg H$_2$O m$^{-2}$ yr$^{-1}$) | 960 | 249 | 422 | 3 | 291 | | | 5 |
| Carbon balance | $GPP$ | $R_a$ | $R_h$ | $W_h$ | | $\Delta W$ | $\Delta C_{soil}$ | |
| (gC m$^{-2}$ yr$^{-1}$ ) | 2336 | 1401 | 664 | 148 | | 64 | 65 | -6 |

– The closure of the carbon balance was also satisfactory, the balance between the gross primary production and the sum of carbon stock changes in biomass and soil, harvested carbon plus the ecosystem respiration being less than 0.3%. The mean annual $NEE$ over the period was 266 gC m$^{-2}$ and is partitioned among three parts, the amount of carbon exported by harvesting, $W_h$ and the net annual increments in biomass, $\Delta W$, and soil organic carbon, $\Delta C_{soil}$.

## 3.2 Parameterisation

The complete list of the parameters of the model is provided in the supplementary material Table S1 together with an appropriate reference. Most of the main parameters of the model have direct observational counterparts and their values were extracted from the literature or open data sources.

Soil.

The soil parameters of the GO+ model — as listed in Table S1 — are essentially functional and not descriptive. The rooting depth, $z_{roots}$, is the depth equivalent of the soil volume affected by the root water uptake. It should not be interpreted as the maximal depth at which roots can be observed — that can be substantially deeper. The parameters $\theta_{FC}$, $\theta_{SAT}$, $\theta_{WP}$, have been estimated by pedotransfer functions from the kinetics of soil humidity retention curves collected over Europe and France (Wosten et al. 1999, Roman-Dobarco et al. 2019) . The parameters are dynamic and depend upon the organic matter content of the soil calculated at a daily resolution. The soil water potential $\psi_{soil}$ (MPa) and hydraulic resistance, $r_{soil}$ (Eq.19) are calculated from the soil texture and water content following Van Genuchten (1980) with soil texture - dependent parameters estimated using the approach developed in Ghanbarian-Alavijeh et al. (2019).

Initialisation of the soil carbon stock is prescribed by the user and may correspond either to observed values or to steady state values simulated by model spin-up. The organic layers above-ground, eventually including coarse woody debris, are conceptually included in the $DPM$ and $RPM$ fractions of the model. They are not separated from the mineral soil (layers $A, B, C$) for the calculations of energy and water exchange. Each type of plant material either foliage, branches, stem, roots,





perennial part of the understorey etc. is characterised by a specific prescribed composition of decomposable and resistant plant material for each species considered.

Canopy layers

The model parameters generally refer to the entire vegetation layer, i.e. to either the trees, understorey or soil layers. This is
certainly the case for carbon metabolism parameters related to the vegetation respiration or photosynthesis. The main model assumption concerns the horizontal homogeneity of canopy layers and implies within-population variations in canopy parameters are ignored. Ideally, the optical and radiative parameters of the canopy layers will have been estimated from data observed either at leaf or canopy levels, in situ or remotely (Hassika et al. 1997, Breda 2003). The stomatal conductance model should be parameterised from measurements upscaled to the canopy level ( Granier and Loustau 1994, Granier et al. 2000a, Rayment et
al. 2000, 2002). The response functions have been thus parameterised based upon the data available from Granier and Loustau (1994), Delzon and Loustau, (2005) and Granier et al. (2000a) for pine, Granier et al. (2000a) for beech or Van Wijk et al. (2000) for Douglas fir, Medlyn et al. (2001) for the $CO_2$ response. For example the functions used for pine are as follows:

$$f_{SW} = \frac{SW_{a,c}}{SW_{a,c} + k_{SW}}$$

$$f_{\delta e} = k_{e1} \times \delta e_{w,c}^{-k_{e2}}$$

$$f_{\psi} = \frac{1}{1 + (\frac{-\psi_c}{k_{\psi 1}})^{k_{\psi 2}}}$$

$$f_{CO2} = (1 - (1 - k_{CO2}) \times (\frac{CO_2}{350} - 1))$$  (35)

The bulk root-to-leaf tree hydraulic resistance is modelled empirically from literature data documenting combined measurements of transpiration or sap flow and soil and leaf water potential (e.g. Loustau et al. 1990, 1996, Delzon et al. 2003, Granier et al. 2000). The parameters used for describing the rainfall interception and its retention by the canopy layers were extracted
from field data analysis (see discussion on parameters estimates in Muzylo et al. 2009). In the version 3.0 of the model, the value of the fraction of carbon allocated to growth is identical for all biomass parts and fixed at 0.28 (Penning de Vries, 1974). The phenology model of understorey vegetation is based on the understorey at Le Bray and other sites (Loustau and Cochard, 1991, Moreaux, 2012).

The allometric parameters used for allocating the net carbon produced to the different tree parts are derived from sets of allo-
metric equations published in the literature and commonly available for the main commercial tree species. Most of them are robust enough to be applied to a range of soil, climate and management conditions (e.g., Gholz et al., 1979, Wutzler et al. 2007, Shaiek et al. 2011). The leaf area index is calculated from the total foliage biomass using the specific leaf area as follows;

$$LAI_T = W_{f,T} \times SLA_T \times \xi$$  (36)

$$LAI_U = W_{f,U} \times SLA_U$$  (37)

where $\xi$ is the leaf area to LAI ratio.





# 4 Sensitivity and uncertainty assessments

We focused the sensitivity analysis presented below on the LE Bray site that was monitored from 1987 to 2010. It is a well-documented site and the data meet our objective, which was to verify the consistency of the model rather than to investigate geographical or climate variations of ecosystem functioning. A one-at-a-time (OAT) sensitivity test was carried out considering first the model parameters, and second the climate variables. This analysis aimed to: (i) check the consistency of the model behaviour in response to step changes in its main parameters and meteorological forcing variables; (ii) investigate possible interactions between the model sensitivity and climate; and (iii) compare the short-term to the long-term sensitivities of the model.

We used the time series of meteorological data interpolated across the SAFRAN grid from 1970 (planting) to 2010 (final cut) as well as the parameters related to soil characteristics and the forest tree stand (stocking density, soil preparation, understorey removal, thinning and harvest). The data used are available at the ISI-MIP project web site and the Fluxnet database (http://dx.doi.org/10.18140/FLX/1440163). We analysed the sensitivity at three temporal resolutions, hourly, annual and full rotation (40 years). The parameters' mean values, the meteorological and soil datasets as well as initial stand conditions were all taken from the European data cluster database (http://gaia.agraria.unitus.it/home). The sensitivity index of a given model variable Y to a parameter — or variable — k was calculated as its response to a step variation of k as:

$$I_k = \frac{Y(1.1 \times k_{ref}) - Y(0.9 \times k_{ref})}{2} \tag{38}$$

where $k_{ref}$ is the reference value for the parameter. All the other parameters are fixed at a nominal value (mean or final value). This index is the variation of $Y$ in response to a 10% step change in $k$. To some extent, the $I_k$ values are more meaningful than mean — or sigma — normalised indices, especially for variables that may take values close to zero such as, NEE. The relative values were also computed for easing the comparison between parameters across Figs. 7–9. The relative values

$$I_{k,rel} = \frac{I_k}{Y(k_{ref})}$$

were also computed for easing the comparison between parameters.

## 4.1 Sensitivity assessment: model parameters

The sensitivity analysis of model parameters was restricted to a subset of the 28 parameters with the aim of giving a general assessment of the model behaviour in response to its parameter variations. The parameters considered are distributed among six groups related to different processes or canopy layers: structure and allometry, phenology, radiation transfer, soil parameters, tree physiological parameters and understorey physiological parameters. They cover, therefore, the main processes accounted for by the model: leaf unfolding, growth and senescence, radiation and energy balances, hydrology, photosynthesis, respiration, soil carbon balance, tree growth and production. The parameters are assumed independent, i.e. their effect on output variables is approximately additive. The effects of factor interactions on the output variance are neglected with OAT methods, which are therefore only applicable to strictly additive models (Campolongo and Saltelli, 1997). The Y output variables describe





the energy balance, water and carbon cycles, carbon balance and tree canopy growth and structure, resulting in a total of 21 variables. The sensitivities of variables related to canopy growth and structure are shown only for the entire rotation. The hour and year sensitivities were calculated separately for a wet year and a dry year, 1994 and 2005, which received precipitation of 1271 and 681 mm, respectively. For each year, the same set of parameter values and the same initial soil and stand conditions

were used. Since the hour and year sensitivities provided essentially the same sensitivity profile, only the year sensitivity is shown. The Figs. 7 –9 show the relative sensitivity index for selected parameters whereas the complete table of results are given in supplementary material Figs S1-S3.

    Independent of the annual climate, the most influential groups of parameters were first the soil characteristics (the rooting depth and water contents at field capacity and at wilting point); second, the tree canopy physiological parameters and the spe-

cific leaf area of both tree and understorey foliage. The model parameters related to the radiation transfer, $\alpha_{soil}, k_{b,T}, k_{d,T}, \rho_{T,f}$, and phenology, $BB_T$ and $GD_U$ had a lesser influence. The relative sensitivity of output variables increased according to their position in the process chain, the sensitivity of end variables, e.g., $NEE$, being the largest and reaching 14 and 45% for the 1994 (wet ) and 2005 (dry) year, respectively. The higher sensitivity of $NEE$ is not unexpected since this variable results from the whole chain of processes related to the energy and carbon fluxes: its sensitivity profile accumulates the impacts of param-

eter changes on the canopy photosynthesis $GPP$ , and autotrophic respiration,$R_a$. Conversely, the energy balance and water balance components, $Rn$, $H$, $\lambda E$ and runoff, $D$, exhibited a low relative sensitivity, their relative change being close from 0.04 and exceeding 0.10 only for the soil water content at field capacity, $\theta_{FC}$, that has enhanced the soil water storage and mitigated the water stress impact. Comparatively, the model outputs were more dramatically affected by the changes in the wilting point $\theta_{WP}$ because of its larger impact on the soil pressure head, water potential and hydraulic resistance and in turn on leaf water

potential (Eq. 21), canopy stomatal conductance (Eq. 13), photosynthesis (Eq. 22), stress index and allocation (Eq. 29, see also Table S5 the impact on understorey). The sensitivity of the carbon balance components was distributed more evenly among the parameter groups. Comparing the sensitivity of the variables groups between 1994 and 2005 revealed also differences that can be related to the contrasting amount of precipitations and related impacts on soil moisture deficit and plant water stress. The absolute sensitivity was higher for the wet year 1994. This is attributed to the fact that the absolute annual values of most

variables were higher for this year. The higher annual values of most variables for 1994 explains also why, apart from the respiration components $R_a, R_{ECO}$ and $R_h$, the relative sensitivity of output variables almost doubled in 2005. Another noticeable contrast between 1994 and 2005 was the shift of the sensitivity of the carbon balance components $NEE, GPP$ and $NPP$ to the photosynthetic quantum efficiency, $\alpha_T$ and carboxylation efficiency, $V_{cmax}$ , that was prominent in 1994 and minor in 2005. The opposite was observed in 2005 for the soil water content at wilting point whose sensitivity increased from $14.4, 3.4$

and $6.2\%$ in 1994 to $45.5, 8.6$ and $13.7\%$.

    The pattern of the long-term sensitivity evaluated from 1970 to 2010 is shown in Fig. 8 for the "flux" variables and Fig. 9 for the tree growth and "stock" variables. The main features previously shown by the sensitivity analysis on annual values were confirmed except that the impacts of the tree foliage $SLA$ and diffuse light attenuation coefficient $k_d$ were enhanced whereas the understorey related parameters had less influence. The sensitivity of the biomass and soil carbon stocks, $W$ and $C_{soil}$, mean

stem diameter and height reached in 2010, $D_{130}$, $H_c$ and cumulated harvestable production $W_{h,stem}$ was consistent with the



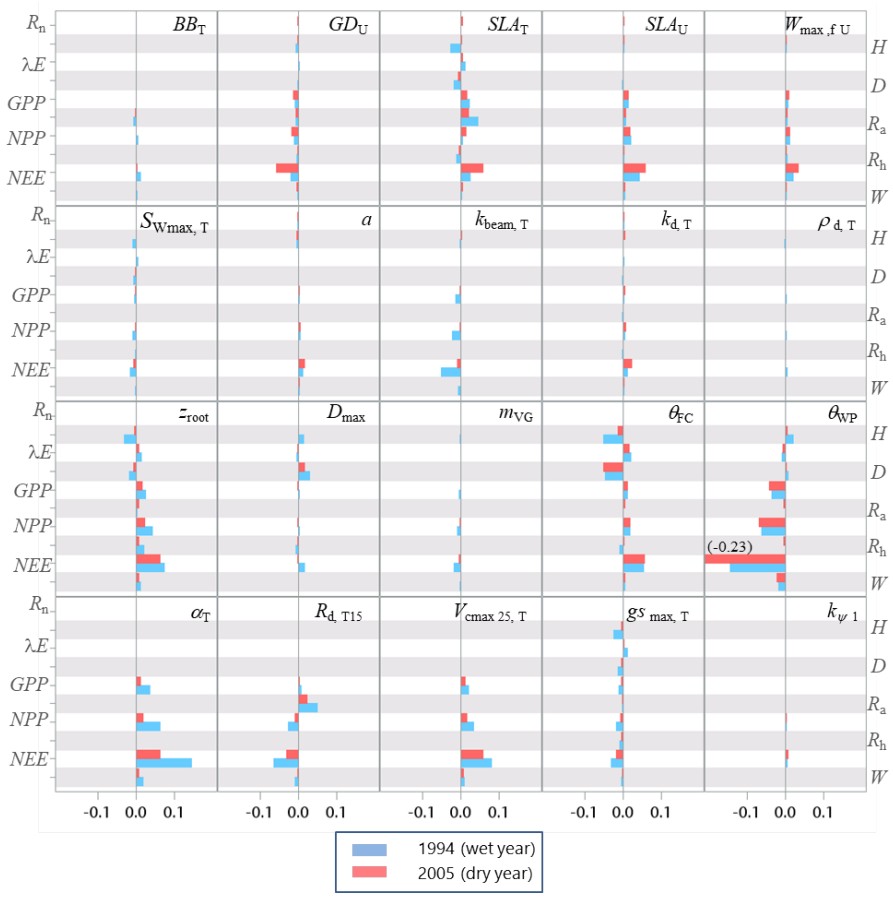

**Figure 7.** Relative sensitivity index values of the main variables related to the energy, water and carbon fluxes to model parameters for the years 1994 (wet, in blue) and 2005 (dry, in red) at the Le Bray site. The horizontal bars in each box gives the relative sensitivity of 10 variables listed along the $y$ axis to the parameters named in the box. A positive value means that the output variable increased in response to an increase in the parameter value.

patterns observed previously on fluxes. The commercial production was the most sensitive to the model parameters $SLA$, $\theta_{WP}$ and to the allometric parameters, $k_{D_{130}1}$, $k_{Istress1}$ and $k_{stem,1}$.

## 4.2 Sensitivity assessment: meteorological variables

The model behaviour in response to variations in meteorological variables was analysed following a similar approach. We con-
5 sidered the following variables: air temperature, atmospheric pressure, precipitation, mean horizontal wind speed, downward shortwave and longwave atmospheric radiation, ambient $CO_2$ concentration and water vapour pressure saturation deficit, and the fraction of diffuse radiation. The air temperature and air vapour saturation deficit were changed by $\pm 1^oC$ and $\pm 200$ Pa, respectively, and other variables were changed by $\pm 10\%$. This analysis was conducted for the same periods as above: the years





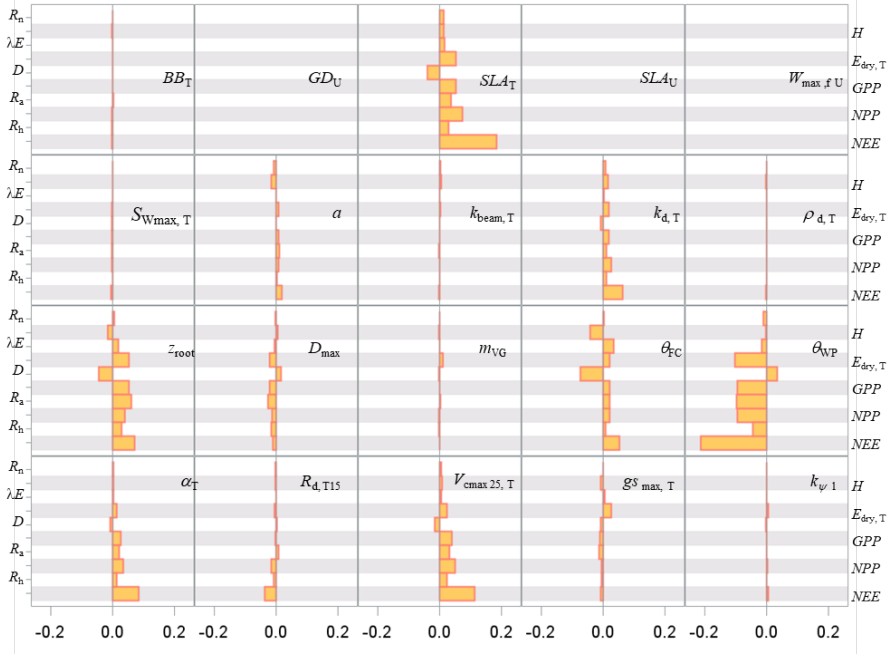

**Figure 8.** Sensitivity index values of the main variables related to the energy, water and carbon fluxes to model parameters over a full rotation (1970-2010) at the Le Bray site.

1994 and 2005 and the 40-year long rotation cycle 1970–2010. The results are presented in Fig. 10 for the annual sensitivity and in the supplementary material (Fig. S4) for the long-term sensitivity. The main conclusions are summarised below. The model overall behaviour was consistent with the current knowledge about canopy responses to climate for the ecosystem considered: a temperate Atlantic coniferous ecosystem growing on a well drained sandy soil for the present case (Granier and

Loustau 1994, Medlyn et al., 2001, 2002, 2003, Davi et al. 2006, Moreaux et al. 2011, 2020). On an annual basis, the energy balance components, $Rn, H, \lambda E$, were mainly affected by incident radiation, $LW \downarrow$ and $SW \downarrow$ and $T_{air}$ whereas the carbon balance variables, $GPP, R_a, NPP, R_h$ and $NEE$ were more sensitive to $CO_2$, $f_{dif}$ and precipitations $Rain$. The negative response of the sensible heat flux $H$ to the air temperature was essentially due to the asymmetric response of $H$ with respect to the sign of $T_s - T_{air}$ that was amplified when $T_s - T_{air}$ was negative. Changes in the air temperature and water vapour

saturation deficit had a negative effect on all variables except the latent heat flux and respiration for the air temperature. It is worth noting that the effects of $CO_2$ and $f_{dif}$ were first to impact $GPP$, then to affect $NPP$ ($= GPP - R_a$), and last $NEE$ ($= NPP - R_h$). The air temperature and incident long-wave radiation also had significant impacts on the respiration terms $R_a$ and $R_h$. The weak response of the carbon processes to a 10% change in $SW \downarrow$ has also been observed, e.g. by Delpierre et al. (2012) under temperate climate was not unexpected since the light is not limiting at this site. The main contrast between

the 1994 and 2005 climates was observed on $NEE$ and $H$, the sensitivity of the former being enhanced in 2005 while $H$ was conversely more sensitive in 1994 . The full rotation sensitivity profile of the "flux" variables (Fig. S4) was identical to



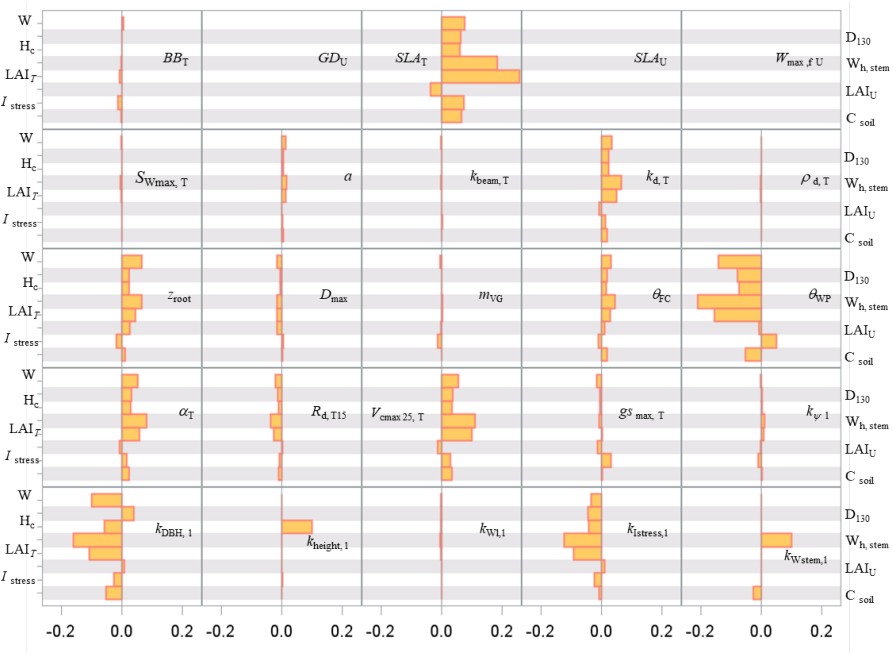

**Figure 9.** Sensitivity index values of the main variables related to the carbon stocks in biomass and soil to model parameters over a full rotation (1970-2010) at the Le Bray site.

the annual sensitivity profile, apart from the biomass and soil respiration, $R_a$ and $R_h$, and consequently $NEE$. In particular, the sensitivity of $R_a$ to the air temperature and longwave incident radiation was positive on an annual basis, as expected from Eqs. (24–26) but became negative over the long rotation. This reversal is induced the long term impacts of atmospheric and soil droughts correlated with the temperature step increase as shown by the enhancement of the stress index in response to temperature (Fig. S5). The temperature step–increase depleted the biomass growth, $W$, and in turn photosynthesis $GPP$ and respiration $R_a$. The same response is shown to the long wave radiation $LW \downarrow$ that increased the water stress index $I_{stress}$ on the long term.

### 4.3 Uncertainty assessment

For assessing the uncertainty of the main variables simulated by GO+, we used a simple Monte Carlo approach where 2500 sets of parameter values were randomly drawn from their distribution range. For each set, the model was run for the year 1994 at Le Bray. Based upon the previous sensitivity analysis, we retained the 14 most sensitive parameters for assessing how errors in parameter values are projected on GO+ output variables. The parameters selected were assumed independent. We are aware this assumption may not hold for biological and physiological parameters but we lack of quantitative relationships that would allow us to link them and define a more sound sampling design. The probability distribution assigned to each parameter was





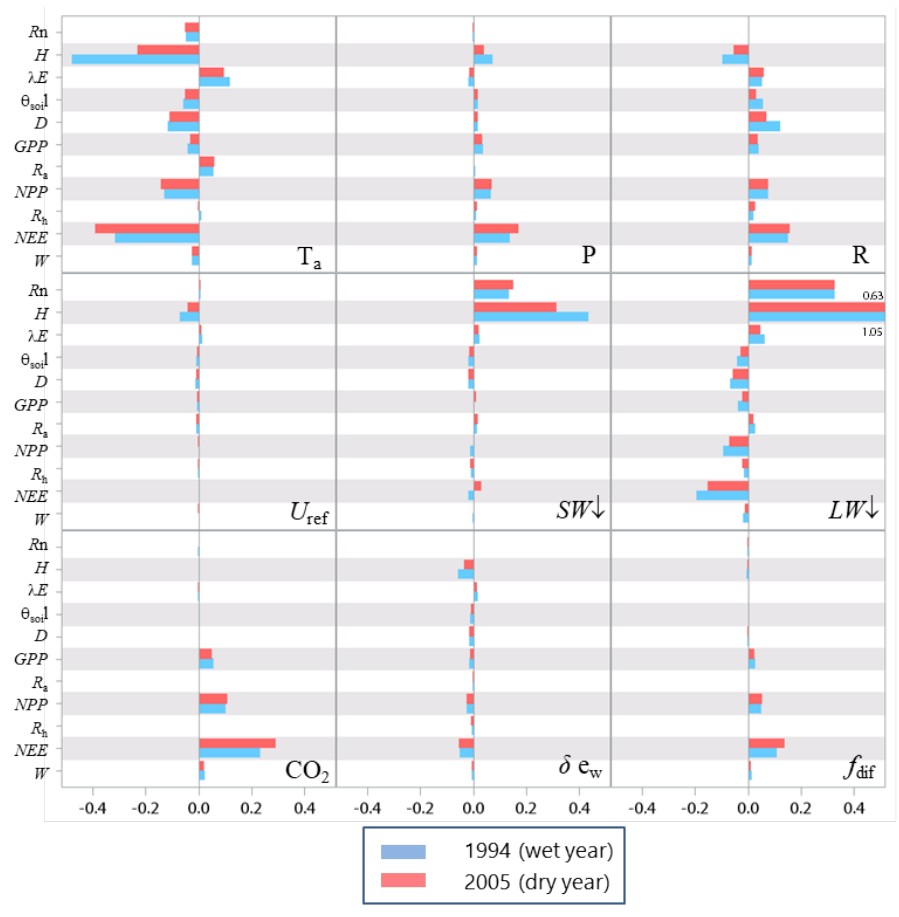

**Figure 10.** Relative sensitivity of the main variables related to fluxes of energy, water and carbon to meteorological variables for a wet (1994) and dry year (2005) at the Le Bray site . The definition of the variables is provided in the Table 1.

by default a normal distribution function (Table 3). The resulting distributions are shown in Figs. 11- 12 for the ecosystem variable values.

The output variables were standardised to their mean value in order to compare the uncertainties among variables and canopy layers. Only the ecosystem variables are shown, the uncertainty in variables referring to canopy and soil layers are

5 reported in the supplementary material (Figs. S6-S8).The uncertainty range of the energy balance components and ecosystem shortwave albedo was relatively small. It was highest for sensible heat flux, $H$ , and lowest for the net radiation, $Rn$. This was attributed to the relative accuracy of the attenuation coefficients in direct and diffuse light that were both measured at this site (Berbigier and Bonnefond, 1995), and the fact that the uncertainty of the longwave emissivity was not considered. In addition, compensation effects between canopy layers and soil might have reduced the range of simulated net radiation. Compensation

10 between layers may also explain the relative precision of the model on the ecosystem albedo because any error on the radiation transfer through the upper canopy will mechanically induce an opposite change in the understorey balance. Indeed, the error





**Table 3.** List of the parameters used for uncertainty propagation in the GO+ model, their reference value and standard deviation.

| Parameter | Symbol | Reference value | Standard deviation | Unit |
|---|---|---|---|---|
| Tree SLA | $SLA_T$ | 6.5 | 0.50 | $m^2 kg\ dm^{-1}$ |
| Understorey SLA | $SLA_U$ | 20 | 1.5 | $m^2 kg\ dm^{-1}$ |
| Heat sum for the tree foliage bud burst | $BB$ | 1400 | 50 | $^oC$day |
| Growth duration of understorey foliage | $GD_{U,f}$ | 130 | 15 | day |
| Maximal understorey foliage biomass | $W_{max,f}$ | 0.25 | 0.03 | kg dm $m^{-2}$ |
| | | | | |
| Canopy extinction coefficient for a beam normal to the surface | $k_{bh,T}$ | 0.33 | 0.02 | |
| Canopy extinction coefficient for diffuse SW radiation | $k_{b,T}$ | 0.467 | 0.03 | |
| | | | | |
| Rooting depth | $z_{root}$ | 0.7 | 0.1 | m |
| Maximal drainage rate | $D_{max}$ | 2.5 | 0.25 | kg $H_2O m^{-2}\ hr^{-1}$ |
| Van Genuchten $m$ | $m_{VG}$ | 0.75 | 0.08 | |
| Water content at field capacity | $\theta_{FC}$ | 205 | 10 | kg $H_2O\ m^{-3}$ soil |
| Water content at wilting point | $\theta_{WP}$ | 65 | 5 | kg $H_2O\ m^{-3}$ soil |
| | | | | |
| Quantum efficiency | $\alpha_T$ | 0.14 | 0.02 | mol $e^- mol^{-1}$photons |
| Foliage mitochondrial respiration at $25^oC$ | $R_{d,T}$ | 8.0E-7 | 1.0E-7 | mol $CO_2 m^{-2}s^{-1}$ |
| | | | | |
| Maximal carboxylation rate at $25^oC$ | $V_{cmax,T}$ | 45.0E-6 | 7.5E-6 | mol $CO_2\ m^{-2}s^{-1}$ |
| Canopy water storage capacity | $S_{wmax,T}$ | 0.25 | 0.03 | kg $H_2O\ m^{-2}$ soil |
| Maximal stomatal conductance | $g_{smax,T}$ | 4.24E-3 | 3.5E-4 | m $s^{-1}$ |

generated on the energy balance components of canopy layers was higher for the understorey and the soil; these were poorly constrained as compared to the tree and ecosystem energy balance. The uncertainty on carbon flux variables was higher than that for the energy balance, especially the net ecosystem exchange, $NEE$, that accumulated the errors generated on both the gross primary production, $GPP$, and the autotrophic, $R_a$, and heterotrophic, $R_h$, respiration components. Our experiment
5    might have exaggerated the error on $NEE$ and $NPP$ since the values of photosynthetic and respiration parameters were drawn independently ignoring the functional link between photosynthesis and respiration. Nevertheless, this relatively large error on $NEE$ will limit the use of its observational counterpart for evaluating the model.

The uncertainty in the annual variation in soil carbon stocks showed contrasting patterns depending on the components considered. The high accuracy on $RPM$ and $DPM$ were to some extent artefacts since the litter biomass was prescribed so
10    that the only error source was caused by the mineralization and humidification processes. Conversely, the $HUM$ component

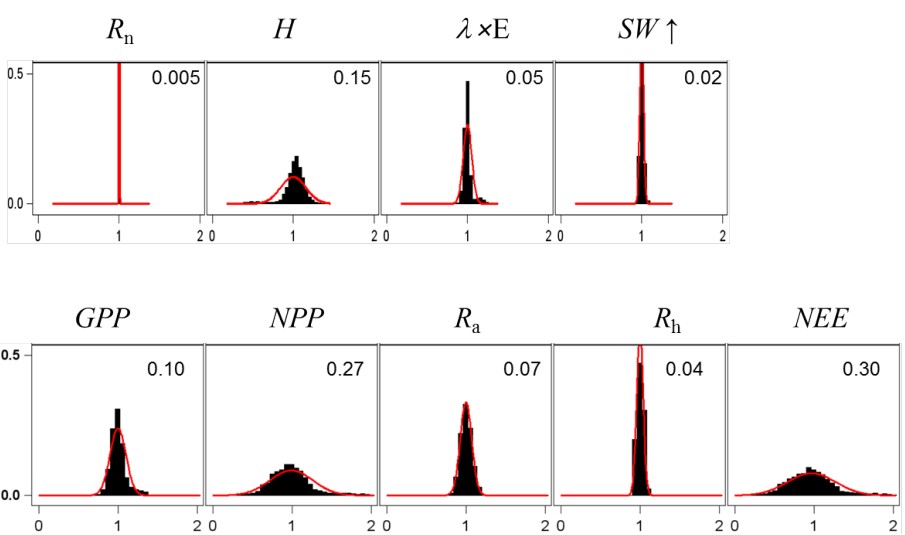

**Figure 11.** Normalised uncertainty on the annual mean values of flux variables predicted by the GO+ model. Each graph shows the distribution of variable values generated from 14 parameter distributions (Table 3). Red curve is the normal distribution fitted and number inset is the standard deviation.

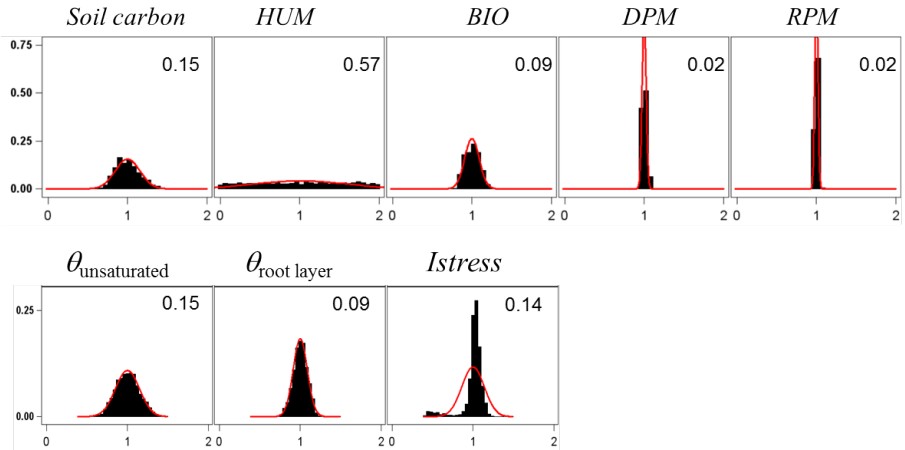

**Figure 12.** Normalised uncertainty on the main soil variables simulated by the GO+ model. Each graph shows the distribution of variable values generated from 14 parameter distributions (Table 3). The red curve is the normal distribution fitted and the numbers inset is the standard deviation.

showed very large uncertainty which was attributed mainly to the fact that uncertainty was related to the stock change that was very small over a year. Overall, the annual change in soil carbon stock was constrained with a standard deviation of 15%. The same magnitude was found for the annual change in the soil water content of the unsaturated layer, whereas the annual change in the total amount of water in the rooted zone was estimated with a precision of 9%. The annual changes in biomass and its





components were not well constrained, its standard deviation exceeding 0.3 in the tree layer and 0.5 in the understorey (Fig. S8). This was not unexpected because the net biomass change is the end result of the whole chain of processes described in the model (phenology, radiation transfer, energy balance and evaporation, photosynthesis, respiration and allocation, and growth), and this chain accumulates their related errors. In addition, the assumption that the parameters are independent might have

inflated the uncertainty in biomass changes.

## 5    Comparison with observed data

Because GO+ encompasses full rotation duration, we were able to test the model against long time series of fluxes and stocks at both daily and annual resolution (Thum et al. 2017). To this end, two types of data were used and the model performance was assessed through two comparisons. First, is a comparison of hourly values of flux data between observed and predicted values.

The second uses annual values of stand growth data. The statistics used are the root mean square error between observations and simulated values, RMSE, the variance fraction explained by the model, $R^2$ , and the systematic and unsystematic model errors which assess the bias and precision of the model respectively (Wallach and Goffinet, 1989).

### 5.1    Data

The time series of daily values of energy water and carbon dioxide fluxes, i.e. net radiation, $Rn$, latent heat flux, $\lambda E$, and

net $CO_2$ fluxes, $NEE$, used in experiment 1) and 3) were obtained from tower stations and taken from the European fluxes database cluster and the Fluxnet Database (URL: http://fluxnet.fluxdata.org) for Douglas-fir sites. The variables used are determined from site measurements of $SW \uparrow, SW \downarrow$ and $LW \uparrow, LW \downarrow$, vertical fluctuations of wind speed, $U$, and fluctuations of $CO_2$ and water vapour concentrations. The values are further processed for quality checking, filtering and gap filling. Unless mentioned, they are all Level-3 type for $Rn$ and Level-4 type for $\lambda E$ and $NEE$. The Level-3 data are standard files provided

by stations. The Level-4 data are filtered, gap-filled using the Marginal Distribution Sampling method (Papale et al. 2006, Moffat et al. 2007) and aggregated at different time resolutions, from half-hourly to yearly. Other variables commonly used for model testing such as ecosystem $GPP$, $RE$ or $NPP$ are derived indirectly from primary measurements. They were, therefore, not used in the model evaluation because that would have introduced redundancy with the test on $NEE$. Table 4 presents the datasets used and their origin. Seven stations were selected because they cover a large part of the geographical range of three

important European commercial tree species and also embrace a wide range of tree stand age.

The data used in experiment 2) is a set of 11 long-term records of stand growth that were mostly taken from the Profound project database (Reyer et al. 2019). The site characteristics and data sources are detailed in the Table S6. In this evaluation, the model performance was assessed using the annual series of stem diameter at 1.3 m height ($D_{130}$ ) and basal area ($BA$) of the tree stands. Three common commercial species are represented: maritime pine, European beech and Douglas fir at dif-

ferent locations across Europe and British Columbia. Various tree ages and thinning regimes are used. We compared the time derivative of the stem mean diameter, $\Delta D_{130}$ (cm yr$^{-1}$), and basal area, $\Delta BA$ (m$^2$ ha$^{-1}$ yr$^{-1}$), that is the cross sectional area of tree stems at 1.3 m height over one hectare. The later can be taken as a proxy for the carbon storage in biomass for which no





**Table 4.** Characteristics of the sites selected for long term series of daily fluxes of net radiation, latent heat and $CO_2$.

| Site name (Fluxnet code) | Lat / Lon (°) | Annual T (°C) Precip.(mm yr$^{-1}$) | Main species | Tree age (yr) | Period | Source |
|---|---|---|---|---|---|---|
| BC-Campbell 49 | 49.86 / 125.33 | 8.4 / 1245 | Douglas fir | 51 | 2000–2010 | (1) |
| BC-Campbell 88 | 49.50 / 124.90 | 9.6 / 1546 | Douglas fir | 14 | 2002–2008 | (2) |
| Hesse (FR-Hes) | 48.67/7.07 | 9.2 / 820 | European beech | 33 | 1996–2010 | (3) |
| Soroe (DK-Sor) | 55.49/11.6 | 8.2 / 660 | European beech | 88 | 1998–2012 | (4) |
| Collelongo (IT-Col) | 41.85/13.59 | 6.3 / 1180 | European beech | 130 | 1997–2014 | (5) |
| Le Bray (FR-LBr) | 44.71 / -0.77 | 13.5/ 930 | maritime pine | 26 | 1996–2008 | (6) |

1-2. Fluxnet,Humphreys et al. (2006)

3.European database, Granier et al. (2000b)

4.European database, Pilegaard et al. (2011)

5.European database, Scartazza et al. (2013)

6.European database, Berbigier et al. (2002)

**Table 5.** Statistics of the model evaluation with daily flux values at 5 sites: $R^2$ and RMSE.

| | $Rn$ (Watts m$^{-2}$) | | $\lambda E$ (Watts m$^{-2}$) | | $NEE$ (gC day$^{-1}$ m$^{-2}$) | |
|---|---|---|---|---|---|---|
| | $R^2$ | RMSE | $R^2$ | RMSE | $R^2$ | RMSE |
| BC Campbell 49 | 0.97 | 15.7 | 0.76 | 13.9 | 0.67 | 1.7 |
| BC Campbell 88 | 0.95 | 15.6 | 0.67 | 13.3 | 0.25 | 1.8 |
| Collelongo | 0.60 | 54.8 | 0.41 | 41.7 | 0.30 | 5.5 |
| Hesse | 0.75 | 41.4 | 0.70 | 28.2 | 0.56 | 3.0 |
| Soroe | 0.59 | 58.0 | 0.65 | 56.8 | 0.51 | 4.0 |
| Le Bray | 0.61 | 44.3 | 0.26 | 23.0 | 0.22 | 2.9 |

direct measurement method exists. Moreover, compared to flux values determined from turbulent variables, the stem diameter and basal area are measured with a low uncertainty (1–5% error) and cover a wide range of climatic, soil and management conditions. [t]

## 5.2 Results

5 To assess the overall performance of the model, we need to relate the RMSE and its systematic and unsystematic components (Table 5) to the model uncertainty in $Rn$, $\lambda E$ and $NEE$. The model errors were larger than the respective uncertainty of the three variables calculated in the previous section. Indeed, our uncertainty analysis used parameter values that had been obtained from local measurements — no site calibration was carried out in this comparison, i.e., a single set of parameters was applied





**Table 6.** (continued). Statistics of the model evaluation with daily flux values at 5 sites: systematic and unsystematic errors.

|  | $Rn$ (W m$^{-2}$) | | $\lambda E$ (W m$^{-2}$) | | $NEE$ (gC d$^{-1}$ m$^{-2}$) | |
| --- | --- | --- | --- | --- | --- | --- |
|  | s | u | s | u | s | u |
| BC Campbell 49 | 3.8 | 13.1 | 8.5 | 13.5 | 1.2 | 1.7 |
| BC Campbell 88 | 2.8 | 15.6 | 6.2 | 13.3 | 1.6 | 1.8 |
| Collelongo | 13.3 | 53.1 | 8.4 | 41.7 | 0.7 | 5.5 |
| Hesse | 3.3 | 36.4 | 17.0 | 28.2 | 0.2 | 2.9 |
| Soroe | 12.6 | 55.4. | 43.3 | 56.8 | 1.8 | 4.0 |
| Le Bray | 18.0 | 41.0 | 22.1 | 23.0 | 0.6 | |

s: systematic error; u: unsystematic error;

to every species, ignoring the acclimation and plasticity of most vegetation traits (Bloomfield et al. 2018). It shows that the model itself introduces a substantial epistemic error in addition to the uncertainty linked to the parameter values. The model error was smallest for $Rn$ and largest for $H$ (not shown), $\lambda E$ and $NEE$. The error on $H$ may be due to the model making the approximation that the aerodynamic resistance to heat transfer can ignore stability corrections. The $NEE$ predictions might be

affected by the simplifications made to the timing of secondary and primary growth of trees and related respiration. In addition, the model represents the source/sink activity and not the transport of water, or $CO_2$ to the reference level. Such a transfer is included in the flux values measured at ecosystem stations and adds a substantial random noise to flux values. The random errors tend to cancel out at larger time spans which can explain partly why the enhancement of the variance explained ($R^2$) at longer time scales (Tables 7 and S7).

The random errors tend to cancel out at larger time spans, which can partly explain the enhancement of the variance explained ($R^2$) at longer time scales (Tables 7 and S7). This is consistent with the relatively small systematic error which indicates that the model predictions have a moderate bias as compared with their precision. Testing the model predictions against observed values at increasing time integrals from an hour to 365 days, we observed that the variance fraction of $Rn$ , $NEE$ and $E$ explained by the model increased with the time span until the (90–day) season-length and then drops at longer time spans (for

example the Hesse site in Table 6 and additional sites in Table S7.

The sources of error are multiple and it should be noted that the data themselves are subject to measurement and calculation errors currently assumed to lie within 10–15% of the daily values used. The meteorological data used may also be a source of errors, i.e., at Le Bray where they were interpolated from the main French national meteorological network. Second, the fact that long time series of variables were used for this evaluation exercise makes the model results affected not only by possible

errors and approximations in processes directly involved in the energy balance and carbon cycle, but also by possible faults affecting the processes describing the vegetation dynamics, i.e., phenology, carbon allocation, tree growth, mortality, forest operations or soil carbon. For this evaluation, the model was run actually from the start to the end of the decadal time series





**Table 7.** Variance fraction $R^2$ of $Rn, \lambda E$ and $NEE$ explained by the model at time spans from 1 to 365 days for the Fr-Hes data set. Numbers in brackets in the last column are the number of observations

| Time span (days) | $\lambda E$ | $Rn$ | $NEE$ | $n$ |
|---|---|---|---|---|
| 1 | 0.70 | 0.75 | 0.57 | (4414) |
| 5 | 0.82 | 0.90 | 0.70 | (878) |
| 10 | 0.85 | 0.93 | 0.74 | (433) |
| 30 | 0.88 | 0.95 | 0.80 | (146) |
| 90 | 0.89 | 0.97 | 0.79 | (50) |
| 182 | 0.83 | 0.95 | 0.31 | (23) |
| 365 | 0.28 | 0.42 | 0.18 | (12) |

without recalibration. This is in particular the case of the Le Bray site where simulations were initiated in 1984 and run until 2000.

The evaluation of the model by comparing its output with long term inventory records reveals that the predictions are relatively close to the observed values, both in terms of accuracy and precision (Fig. 13). Since only few data were available

from inventories, we pooled together in this figure the results of the 11 sites analysed. The data observed are prone to smaller errors (typically 5%) than the previous flux data but the information about the station characteristics and meteorological data used for modelling is more uncertain. This uncertainty is because reliable series of meteorological data, i.e., measured on site, are not available and the information on soil characteristics can be poor. In addition and apart from the Le Bray site, we had only vague information about the criteria used to select which trees are thinned. We used the annual increment rather than the annual

raw values of $D_{130}$ and $BA$ because the latter are actually cumulative variables including large temporal autocorrelation. The predicted $\Delta BA$ were close to the observed ones in general, with most values being positive but close to zero. Interestingly, the model accuracy was mainly constant along all values ranging from -7 to +5 m$^{-2}$ ha$^{-1}$ yr$^{-1}$ . The predicted $\Delta D_{130}$ was satisfactorily simulated by GO+. Given these uncertainties and the fact that no site-specific calibration of parameters was used, the evaluation test shows that the model departs only slightly from measurements on average, with relatively small biases. Its

performance was similar across sites despite the range of species, age, management and location covered by the data set.

An interesting product of the Go+ model is its evaluation based upon simultaneous values observed and predicted of several variables over decades-long time series. To our knowledge few models have been tested using long-term sets of multiple variables. Figure 14 shows an example for the Collelongo broadleaf forest, Vancouver Island Douglas Fir stand and the Le Bray coniferous forest. In Collelongo and Le Bray, 20– and 25–year long inventories of tree diameters were available, respectively.

A series of soil water stock — or groundwater level — and flux measurements were also available in the three sites. The comparison shows that the long-term trajectory of energy, water and carbon fluxes as well as soil water, tree growth and $LAI$ were captured without significant bias by the model for a period marked by severe droughts (2002 and 2005), a heatwave



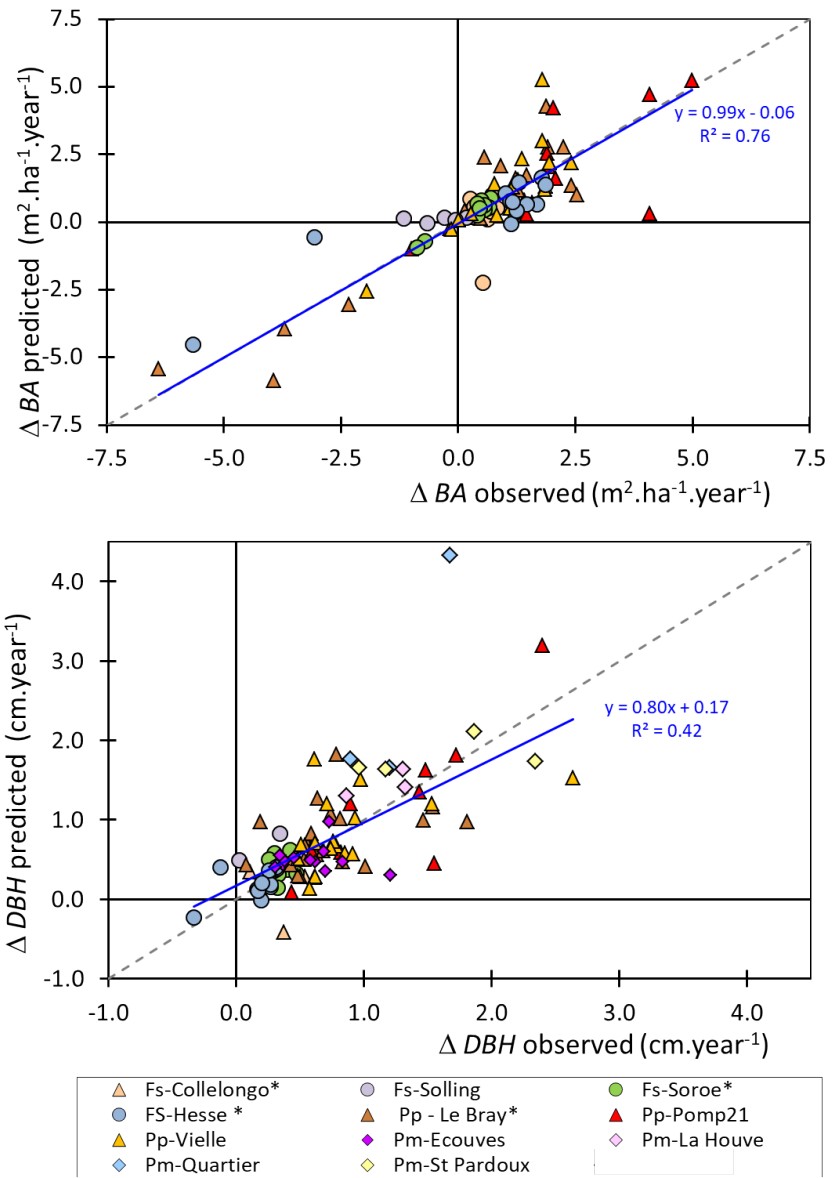

**Figure 13.** Predicted versus observed values of annual increment in basal area, $\Delta BA$, upper diagram, and stem mean diameter, $\Delta D_{130}$ (lower diagram) for sites described in Table S6 of the supplementary material. The sites used for both flux and inventory data are annotated with a star (*). The 1:1 line (dashed line) and linear regression (blue) are shown.

(2003), several thinnings (1992, 1997, 2005 at Le Bray) and storm damage (at Le Bray 20% of trees suffered windthrow in December 1999). Some inconsistencies are also evident, such as the overestimation of respiration and primary production at Collelongo, that may be related to $LAI$ overestimation. The behaviour of the soil moisture predicted at the Douglas Fir site





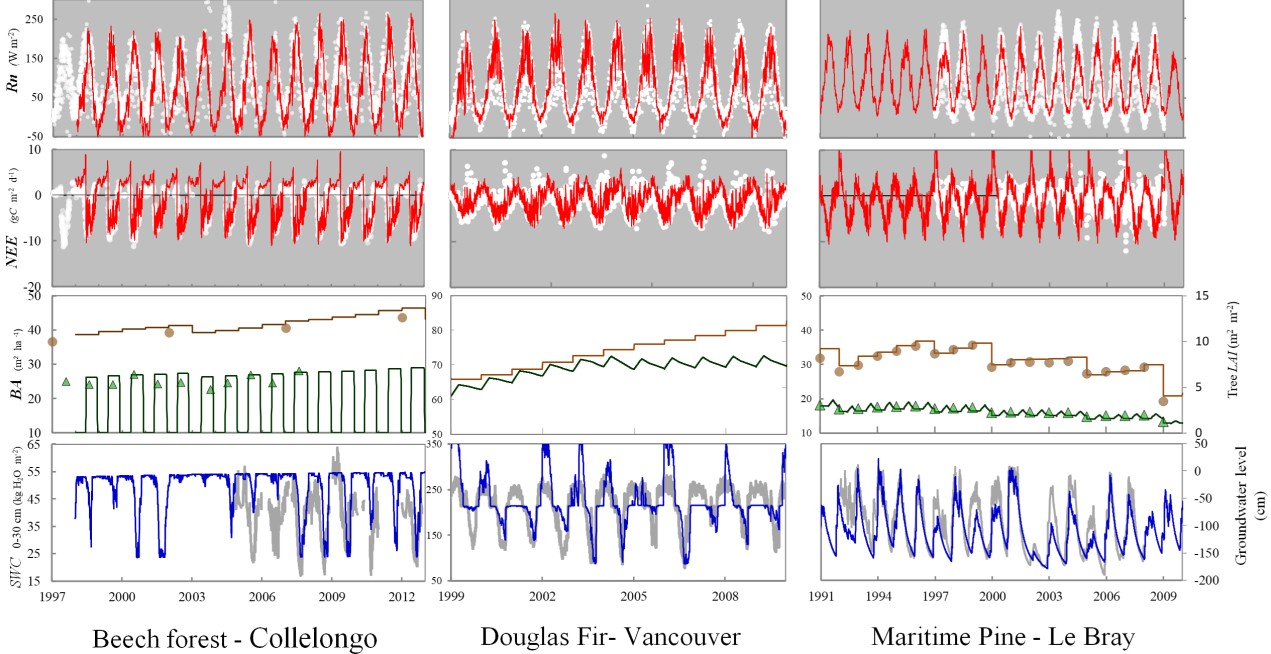

Beech forest - Collelongo    Douglas Fir- Vancouver    Maritime Pine - Le Bray

**Figure 14.** Time series of net radiation and $CO_2$ fluxes observed (white dots and grey lines) and predicted (heavy lines), top diagrams, together with stand basal area, leaf area index and soil water stock or groundwater level, bottom diagrams. Left: Collelongo European beech forest. Centre: Douglas Fir stand in British Columbia. Right: Le Bray pine forest. Source of the data used are detailed in Table 4. The model was initiated in 1997 for the Collelongo experiment, 1998 in the Douglas Fir and 1987 for the Le Bray experiment. The soil water content in the top 30 cm or 60 cm observed at the Collelongo or Douglas Fir sites (left axis of bottom diagrams) are compared with $SWC$ in the rooted zone simulated by GO+. At the Le Bray site, the groundwater level is compared for the 1994–2008 period (right axis).

is also challenged by the observations when soil becomes close from saturation. Because we did not calibrate the parameters for every site, these discrepancies are mainly caused by errors in the values of influential parameters such as the soil depth and hydraulic parameters, the leaf mass-to-area ratio or the root-shoot ratio. The comparison of multiple variables between observed and predicted values also reveals that the performance of the model was clearly affected by the quality of observed

5  flux data. At Le Bray, the flux values were more scattered after 2003 due to a change in instrumentation (closed-path analyser until 2003; open-path from 2004 onward) and related quality assessment criteria: the $R^2$ of the predicted versus observed values of $NEE$ was 0.36 for the period 1997–2003 but dropped to 0.22 when calculated for the entire period 1997–2008. Testing the model against data of different quality levels also produced substantial differences, up to 0.15, in the calculated value of $R^2$.





## 6   Discussion

Essentially, the GO+ model brings together robust representations of canopy and carbon cycle processes that can be evaluated straightforwardly against observed data. From a biogeochemical point of view it offers three main innovations: (i) GO+ explicitly links the stomatal functioning of the tree canopy to the leaf water potential and plant hydraulics; (ii) it allows us

to connect fast biophysical and biogeochemical processes to slower plant growth and soil carbon transformation processes; and (iii) provides for a large range of options in specifying management operations and harvest exportation. In this section, we first discuss these three points and further model specificities. We then return to a discussion of model performance. First, the tree hydraulics model accounts for the effect of the mean tree height and therefore reflects the effects of age on the leaf water potential and stomatal conductance (Delzon et al. 2003). The hydraulic scheme is kept as simple as possible allowing the

description of leaf water potential to be calculated dynamically as a function of transpiration and soil water. The water potential function of canopy stomatal conductance (Eq. 35) is close to the Mencuccini et al. (2015) model, the stomatal closure being smoother in our case. The GO+ stomatal conductance model includes three essential features of the soil-to-leaf water transport, i.e., (i) the soil water potential and conductance dependencies on water content (Eqs. S16 and 22), (ii) the relationship between the tree hydraulic conductance and tree height (Eq. 20), and (iii) its capacitance in relation to the total biomass. We think it is a

satisfying compromise between more sophisticated models, that would be difficult to parameterize and calibrate at large spatial scales, and the need for describing the temporal fluctuations of leaf water potential and related effects on stomatal conductance. Second, the GO+ the net primary production allocation scheme among individual trees and within tree parts satisfies the need to simulate realistically the tree stand size distribution and the harvested wood product categories. This is required for coupling GO+ to models of wood-product life cycles and thus route raw harvest products into a range of product categories, namely:

pulp, biofuel, industrial products, furniture and construction (Pichancourt et al. 2018). The allocation scheme is sensitive to the environmental stresses and management, and satisfies the mass conservation principle. Although simple, this allocation scheme has proven its ability to realistically simulate the dynamics of size distribution in mono-specific stands where the selection of trees thinned is crucial (not shown). It summarises the end result of the carbon transfer processes within a tree. A mechanistic simulation of carbohydrate transport within trees at large spatial scales is still beyond computational capacity and constitutes

a research challenge (Mencuccini et al. 2015).The inclusion of a species-specific set of allometric equations is therefore a trade-off allowing us to constrain the growth allocation within trees. It is parsimonious in terms of parameters, yet confers to GO+ a capacity to account for a wide variety of forest tree species. Moreover, tree allometric equations and parameter values are available for the main tree species of the tropical, temperate and boreal zones (Chave et al. 2014, Forrester et al. 2017). In addition, the prediction of tree size allows the assessment of model performance with data covering a range of temporal

scales from hourly to the complete forest rotation time (Thum et al. 2017) as illustrated by its evaluation at the Collelongo and Le Bray sites. The dynamic allocation scheme implemented in the understorey vegetation results in a temporal dynamics that is consistent with our current understanding of understorey vegetation growth in managed stands, but could not be evaluated yet at a large spatial scale because of data paucity. Unfortunately, long time series of the annual individual tree growth for the entire population of trees are still rare and difficult to obtain.





Third, all possible current practices of forest management are implemented in the GO+ v3.0 code and can be combined in a wealth of forest management options, including: the drainage (not shown) and mechanical preparation of soil, control of vegetation, thinning, coppicing and clear-cutting of the tree stand. The model is being further developed to allow it to simulate tree stands that are not even-aged. Other process-based models also account for a variety of forestry practices (ORCHIDEE-FM,

Bellassen et al. 2011, Reyer et al. 2014, ), but rarely with documented impacts on soil carbon or how the carbon is apportioned to the different harvested products. Regarding the soil carbon, the Roth–C model and its subsequent development, ECOSSE, had been proven useful and relatively accurate for estimating the dynamics of carbon in the top 1.0 m of the soil following afforestation (Romanya et al. 2000, Dondini et al. 2015) or in temperate forests under different climate change scenarios (Smith et al. 2006). The adaptation of the ROTH -C model to include the effects of soil mechanical disturbances, as implemented in the

GO+ model, substantially improves the predictions of soil carbon changes observed following clear-cutting in the pine forests of southwest France. Nevertheless, the model still has to be evaluated at large scale. It is worth noting the GO+ code inherently accounts for the light resource competition between trees and understorey. By construction, the access of canopy layers to light is prioritized from the top to the bottom of the canopy allowing the ground vegetation to respond to thinning and to recover first after clear-cutting. The tree layer subsequently dominates when trees have regrown. This version of the GO+ model also

suffers from a number of limitations. It does not yet include a biogeochemistry module and does not allow us to simulate mixed-stand forests or stands that are not even-aged; this is because so far very few evaluation data sets are available. The uptake of water from the soil is not prioritized, with understorey vegetation and trees having access to the same soil volume and their transpiration being withdrawn from the soil simultaneously. Both this limitation, and the canopy and soil layers homogeneity assumption, could be overcome in subsequent versions of this model through adding a dynamic partitioning of the canopy and

rooted soil. When necessary, the number of layers could also be increased to some extent provided observed data exist to calibrate and validate the canopy structure and simplifying assumptions required. However, the De Pury and Farquhar's radiation and photosynthesis canopy model has proven effective as compared with complex multi-layer models and may well suffice for simulating more complex canopies. Model performance With the above limitations in mind, the overall sensitivity profile of the model is consistent with the current understanding of the role of the different processes involved and their functional hierarchy.

The sensitivity analysis demonstrates an interaction between the sensitivity of variables with the climate — the water holding capacity of soil being limiting under dry conditions, i.e., the year 2005 at Le Bray, and the photosynthetic quantum and carboxylation efficiencies becoming the most influential parameters on wet soil. We have also shown how the time scale modifies the sensitivity profile of model due to the cumulative effects of the fluxes of carbon, energy and water on the stand growth and canopy structure. The GO+ model simulates the seemingly contradictory sensitivity of the autotrophic and heterotrophic

respiration to temperature between the short-term (positive direct thermal) effect and the long-term (neutral or negative) effect, which is linked to reduced productivity (Janssens et al. 2001, Atkin and Tjoelker 2003, Knorr et al. 2005). Although simple, the mechanistic link established between instantaneous canopy processes (radiation and energy balance, transpiration, assimilation and respiration) and longer term processes, such as primary and secondary stem growth, wood production, and soil carbon and water dynamics, allows us to capture dynamically the main trajectory and energy, water and carbon fluxes and stocks over

decades. The main limitation of our model in that respect is the time resolution of the tree growth processes, which does not





account for the seasonality of growth in tree biomass, height and diameter, and may therefore introduce errors, e.g., when predicting the autotrophic respiration at an hourly time-step. This gap may induce some errors for very fast growing species but not for slower growing tree species, as shown in Fig. S3 for the allometric parameters. The sensitivity analysis of GO+ demonstrates that the dynamic representation of stand growth processes is a key feature for capturing the ecosystem behaviour

in the long term. We are aware that the conclusions drawn depend on the sensitivity experiment chosen, in terms of climate, soil, tree species and canopy structure, but we think they will be applicable beyond the specific case examined here, at least for canopies with persistent foliage. Whereas the model performances for energy, water vapour and $CO_2$ flux predictions may compare with other current models (Davi et al. 2005, Collalti et al. 2016, Chen et al. 2016), an essential feature of GO+ is its ability to also capture the long term trends in tree and stand growth and at the same time produce a realistic prediction of the

dynamics of understorey vegetation (not shown) and soil carbon. A shift of influence of meteorological variables at day–month scales to biological factors at yearly resolution and beyond was observed by Delpierre et al. (2012) and Stoy et al. (2005, 2009) for many ecosystems through spectral analysis of $NEE$, $GPP$ and ecosystem respiration sensitivity. This observation suggests the importance of the processes controlling the vegetation dynamics such as phenology, management, growth, and mortality, that are currently described in the GO+ v3.0 model. The accuracy of the model assessed against flux data may appear relatively

poor but it should be noted that a single set of parameter values was used and no site-specific calibration was made. In addition, the most influential site characteristics, the rooting depth and soil hydraulic properties, are unfortunately prone to large errors because of the difficulty in determining them, and their being subject to large spatial variations at the scale of the footprint of flux measurements. Careful examination of the kinetics of predicted and observed flux values reveal that the modelled phenology was not a substantial source of error despite the fact that this process is poorly documented and difficult to parameterise

for species such as European beech or maritime pine. Increasing the complexity of the canopy representation, e.g., by taking into account the heterogeneity of sources and sinks within the vegetation layers, might improve the energy balance and flux modelling (see e.g,. Naudts et al. 2015), but at the expense of the model's ability to simulate sites where no such information is available. Considering the diversity of data sources used for evaluation, the model does not show major discrepancies from observations and performed relatively well, with low biases, at simulating the observed values of atmospheric exchanges, tree

growth, and soil carbon and water stock changes. The satisfactory results obtained from the comparison with long-term historical series of tree and stand growth and soil carbon and water are particularly promising because they confirm the model's ability to capture low frequency variations of forest ecosystem functioning for managed forests and demonstrate its ability to simulate management scenarios under different climate scenarios at regional scale. We could not identify why the GO+ performance for simulating canopy fluxes was so site dependent. It may be in part attributed to the uneven data quality within and between

sites; this may be due to changing instrumentation, data gaps and data processing. Indeed, using data obtained on site (level 3) instead of reconstructed (level 4) quality data produced better performances (not shown). The unique species-specific sets of parameter values used per canopy layer for all sites may also generate deviations from observed values since most influencial plant traits, e.g. $SLA$, $V_{cmax,25}$, $J_{max}$ or $k_N$, exhibit substantial spatial and temporal variations that are not accounted for in our model evaluations (Fajardo and Siefert 2016, Hamada et al. 2016, Bloomfield et al. 2018). Most advanced forest models are

more finely tuned for specific processes, e.g., PnET-BGC for forest hydrology (Gbongo-Gupdawa et al. 2001, Pourmokhtarian





et al. 2012), ANAFORE for cambial growth and carbohydrate storage (Deckmyn et al. 2011), CANOAK, or ORCHIDEE-CAN for light and turbulence attenuation within the canopy (Harley and Baldocchi 1995, Naudts et al. 2015), but are more restricted in terms of temporal scales, process continuity and exhaustiveness. Very few models that can be run over large gridded datasets can implement canopy processes at an hourly time scale: ORCHIDEE-CAN v1.0 (Naudts et al. 2015, Luyssaert et al. 2018),

JULES (Best et al. 2011, Clark et al. 2011) or, optionally, LPJ-Guess v3.0 (Smith et al. 2014). The majority run at daily time scale; that resolution may impair the sensitivity of non-linear processes to climate and $CO_2$ such as photosynthesis, respiration or stomatal function.

## 7   Conclusions

The GO+ model allows us to take a new step forward in developing our understanding of the interactive effects of climate

and management on forest ecosystems. The model integrates biophysical, biogeochemical, growth and management processes across a range of temporal scales from hour to century and beyond. It thus integrates short time scales, at which ecophysiological reactions take place, into the temporal framework at which the ecosystem functions, thereby covering the entire forest rotational cycle. The low biases in the model predictions of the exchanges of energy, water and carbon explains the model's ability to capture the long-term trajectory of tree and understorey growth and production, which is essential for modelling

managed forests. The exhaustive set of forest management options included in GO+ allows a wealth of combinations of forest operations to be implemented and tested. We believe that apart from the nutrient cycles, GO+ includes all the key processes that are needed for understanding the interactions of forest with climate through radiation and the energy, water and carbon cycles, and their impacts on soil and plants, plant growth, phenology and mortality, and wood product exports.

*Code and data availability.* The GO+V3.0 Python code (doi:10.15454/5K9HCS) together with a short user manual and example files (pa-

rameters for sites and species, output files, meteorological data sets) can be downloaded from

https://github.com/DenisLOUSTAU/GOplus_model_INRAE.

The code is also available from the data.inrae.fr repository (https://data.inrae.fr/dataverse/eos) although with less example files. The data used for evaluating GO+ were from the Fluxnet database located at the European Fluxes Database Cluster (http://gaia.agraria.unitus.it/home). The DOI of the data sets of flux sites used are as follows:

– Le Bray:10.18140/FLX/1440163

  – Collelongo: 10.18140/FLX/1440167

  – Soroe: 10.18140/FLX/1440155

The forest inventory data used for Douglas fir and partly maritime Pine were provided by the "GIS" data cooperative (Seynave et al. 2018, https://www6.inra.fr/giscoop), and from the PROFOUND project database (Reyer et al. 2019, http://doi.org/10.5880/PIK.2019.008) for beech

forests (Soroe, Collelongo, Solling).





Table A1: List of the prognostic variables of the GO+ v3.0 model. The table is split among the main processes. The entity subscripts T, U, and S are standing for tree canopy, understorey canopy and soil respectively. The subscript "t" is for individual trees.

| Symbol | Description | Entity (1) | Unit |
|---|---|---|---|
| **1. Radiative balance** | | | |
| $LW \uparrow$ | Upward flux density of longwave radiation | T, U, S | W m$^{-2}$ |
| $SW \uparrow$ | Upward flux density of shortwave radiation | T, U, S | W m$^{-2}$ |
| $SW_a$ | Shortwave radiation absorbed by canopy layers, each separated into shaded and sunlit fractions | T, U, S | W m$^{-2}$ |
| **2.Energy balance** | | | |
| $\lambda E$ | Latent heat flux | T, U, S | W m$^{-2}$ |
| $G$ | Heat storage in the soil | S | W m$^{-2}$ |
| $H_c$ | Sensible heat flux | T, U, S | W m$^{-2}$ |
| $r_{HR,c}$ | Resistance analog to combined heat and radiative transfer | T, U, S | s m$^{-1}$ |
| $r_{R,c}$ | Resistance analog to radiative transfer | T, U, S | s m$^{-1}$ |
| $Rn$ | Net radiation | T, U, S | W m$^{-2}$ |
| $T_{s,c}$ | Surface temperature | T, U, S | °C or K |
| **3. Aerodynamic profiles** | | | |
| $d$ | Zero plane displacement height | T, U, S | m |
| $u*$ | Friction velocity | T, U, S | m s$^{-1}$ |
| $z_0$ | Roughness length for momentum | T, U, S | m |
| **4. Water balance and hydrology** | | | |
| $D$ | Groundwater discharge in absence of evaporation | S | kg H$_2$O m$^{-2}$ hr$^{-1}$ |
| $E_c$ | Evapotranspiration | T, U, S | kg H$_2$O m$^{-2}$ hr$^{-1}$ |
| $E_{wet,c}$ | Evaporation from wet surfaces | T, U, S | kg H$_2$O m$^{-2}$ hr$^{-1}$ |
| $E_{dry,c}$ | Transpiration | T, U, S | kg H$_2$O m$^{-2}$ hr$^{-1}$ |
| $f_{dry,c}$ | Dry fraction of the canopy | T, U, S | - |
| $g_{s,c,h}$ | Surface conductance | T, U, S | m s$^{-1}$ |
| $I_{stress}$ | Stress index[0, 1] | T | - |
| $r_H,c$ | Aerodynamic resistance | T, U, S | m s$^{-1}$ |
| $r_{xyl}$ | Root-to-leaf hydraulic resistance | T | [kg H$_2$O m$^{-2}$ s$^{-1}$ MPa$^{-1}$]$^{-1}$ |





Table A1: (continued) Variables of the GO+ v3.0 model.

| Symbol | Description | Entity | Unit |
|---|---|---|---|
| $r_{soil}$ | Soil hydraulic resistance | S | $[\text{kg H}_2\text{O m}^{-2}\text{ s}^{-1}\text{ MPa}^{-1}]^{-1}$ |
| $z_W$ | Groundwater depth | S | m |
| $\psi_c$ | Leaf water potential (canopy average) | T | MPa |
| $\psi_{soil}$ | Soil water potential (average of the rooted zone) | S | MPa |
| $\theta$ | Water content (split among soil layers A, B, C) | S | $\text{kg H}_2\text{O m}^{-3}$ |
| $\theta_{rootlayer}$ | Water content of the soil root zone | S | $\text{kg H}_2\text{O m}^{-3}$ |

5.Carbon balance

| Symbol | Description | Entity | Unit |
|---|---|---|---|
| $Anet_c$ | Net assimilation (split among sunlit and shaded fractions of foliage) | T, U | $\text{mol CO}_2\text{ m}^{-2}\text{ leaf area s}^{-1}$ |
| $c_c$ | Internal concentration in $CO_2$ | T, U | $\text{mol CO}_2\text{ mol air}^{-1}$ |
| $g_{m,c}$ | Leaf internal resistance to $CO_2$ transfer | T, U | $\text{mol CO}_2\text{ m}^{-2}\text{ s}^{-1}$ |
| $GPP_c$ | Gross primary production | T, U | $\text{gC m}^{-2}\text{ h}r^{-1}$ |
| $NEE$ | Net Ecosystem $CO_2$ exchange | T, U | $\text{gC m}^{-2}\text{ hr}^{-1}$ |
| $NPP_c$ | Net primary production | T, U | $\text{gC m}^{-2}\text{ hr}^{-1}$ |
| $R_{d,c}$ | Mitochondrial respiration during day | T, U | $\text{mol CO}_2\text{ m}^{-2}\text{ leaf area s}^{-1}$ |
| $R_{ECO}$ | Ecosystem respiration | E | $\text{gC m}^{-2}\text{ hr}^{-1}$ |
| $R_a$ | Autotrophic (plant) respiration | T, U | $\text{gC m}^{-2}\text{ hr}^{-1}$ |
| $R_{g,c}$ | Growth respiration | T, U | $\text{gC m}^{-2}\text{ hr}^{-1}$ |
| $R_{m,c}$ | Maintenance respiration | T, U | $\text{gC m}^{-2}\text{ hr}^{-1}$ |
| $W_T$ | Carbon stock in tree biomass (split into stem, branch, leaves, stump, coarse, small and fine roots) | T, t | $\text{gC m}^{-2}$ or gC individual$^{-1}$ |
| $W_U$ | Carbon stock in understorey biomass (split into leaves, perennial part, roots) | U | $\text{kg DM m}^{-2}$ |

6. Soil carbon

| Symbol | Description | Entity | Unit |
|---|---|---|---|
| $BIO$ | Carbon stock in soil: biological fraction | S | $\text{gC m}^{-2}$ |
| $C_{soil}$ | Total stock of carbon in soil | S | $\text{gC m}^{-2}$ |
| $DPM$ | Carbon stock in soil: decomposable fraction | S | $\text{gC m}^{-2}$ |
| $HUM$ | Carbon stock in soil: humified fraction | S | $\text{gC m}^{-2}$ |
| $RPM$ | Carbon stock in soil: resistant fraction | S | $\text{gC m}^{-2}$ |





Table A1: (continued) Variables of the GO+ v3.0 model.

| Symbol | Description | Entity | Unit |
|---|---|---|---|
| $R_h$ | Soil microbial respiration (or heterotrophic respiration) | S | $gC\ m^{-2}\ hr^{-1}$ |

**7. Canopy structure and growth**

| Symbol | Description | Entity | Unit |
|---|---|---|---|
| $A_l$ | Leaf area | t | $m^{-2}\ tree^{-1}$ |
| $BA$ | Basal area (projected cross sectional area of tree stems) | T | $m^{-2}\ m^{-2}$ |
| $D_{130}$ | Tree diameter at $z$=1.3 m above-ground | T, t | m |
| $DOY_B$ | Budburst date | T, U | day of year |
| $DOY_S$ | Senescence date | T, U | day of year |
| $H_c$ | Canopy height | T, t, U | m |
| $LAI_c$ | Canopy leaf area index | T, t, U, E | $m^{-2}\ m^{-2}$ |
| $SD$ | Stocking density | T, U | $m^{-2}$ |
| $V$ | Stem volume | T | $m^{-2}\ m^{-2}$ |
| $WAI$ | Branch and stem area index | T, t | $m^{-2}\ m^{-2}$ |
| $\Delta H_c$ | Annual increment in height | T, t, U | $m\ yr^{-1}$ |
| $\Delta D_{130}$ | Annual increment in stem diameter | T, t | $m\ yr^{-1}$ |

**8. Harvest and mortality**

| Symbol | Description | Entity | Unit |
|---|---|---|---|
| $D_{130,h}$ | Stem diameter at 1.3m height of trees harvested | t | m |
| $h_h$ | Stem height of trees harvested | t | m |
| $M$ | Mortality (harvest excluded) | T | number of trees.$m^{-2}\ yr^{-1}$ |
| $S_{stem}$ | Stem senescence | T, t | $kg\ DM\ m^{-2}\ year^{-1}$ |
| $S_r$ | Root senescence | T, t | $kg\ DM\ m^{-2}\ year^{-1}$ |
| $S_{br}$ | Branch senescence | T, t | $kg\ DM\ .m^{-2}\ year^{-1}$ |
| $T_h$ | Trees harvested | T | number of trees $m^{-2}\ year^{-1}$ |
| $W_h$ | Carbon exported by harvest (split into stem, branch, leaves, stump) | T, t | $gC\ m^{-2}\ year^{-1}$ |





*Author contributions.* VM developed different modules of the model, the understorey, soil and part of the management and she synthesized and reviewed the final version of the manuscript. DA, AB , CC, SM, D P-D, RV, and RA co-developed the model, its adaptation to different species, different modules and parts of the sensitivity and uncertainly analysis. CM did implement the code over gridded datasets using parallel computing. VB, ATB, J-MB, AG, CJ, BL, GM, MN and KP provided the data sets used about the soil carbon, tree and understorey

phenology, forest inventories and canopy fluxes of energy and $CO_2$ and contributed to data analysis and model evaluation. OP and OR were leading the projects supporting the development of the version 3.0 GO+. They supervised the manuscript structure, content and lay-out. DL developed preliminary versions of the model, made the numerical experiments described in the verification and model evaluation sections and wrote the first version of the manuscript article.

*Competing interests.* The authors declare that they have no conflict of interest.

*Acknowledgements.* The datasets used for evaluating the GO+ model with stand growth data and tree species phenology were provided by C. Meredieu (INRAE UEFP0570) for maritime pine permanent plot, the 'GIS Coopérative de données' https://www6.inra.fr/giscoop/ (Douglas fir and maritime pine data sets), French ICP-Forest network (French part of ICP-Forests Level II network –Renecofor–) and Christopher Reyer (PIK - Potsdam) and the ISI-MIP project for the beech data sets. The data sets of flux measurements were downloaded from the European Data Cluster and Fluxnet (La Thuile 2015 dataset). During their work on this article, D P-D and RV were supported by the ANR

projects ORACLE (10-CEPL-0011) and MACCAC (13-AGRO-0005) and DA by the ADEME project "Evafora" (13-60-C0092). VM was supported by the European project RINGO (730944).The authors thank J.H.C. Gash for reviewing the English and style of the manuscript.



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
