# Peer review of "Energy, water and carbon exchanges in managed forest ecosystems : description, sensitivity analysis and evaluation of the INRAE GO+ model, version 3.0."

_Geoscientific Model Development, 2020_

## Referee Comment (RC1) · Anonymous Referee #1 · 18 Jun 2020

**Overall comments:**

This paper is a very detailed and thorough description paper of the newly developed model GO+ version 3, which is a model for simulating carbon, energy and water fluxes in temperate managed forest ecosystems. The model description is detailed, the sensitivity analysis and the model evaluation is extensive, and the discussion of the usefulness and limitation of the model is also adequate. I don't have major issues with the overall content of the paper. I think the paper and the model are both valuable additions to the community, and the paper will be a great contribution to GMD. Below I list several relatively major comments and my detailed comments are available thereafter.

**Major comments:**

The authors claim that the model is novel in that, it combines biophysical and biogeochemical processes of natural vegetation dynamics with different representations of forest management, and thus the model allows the explicit simulation of both short- and long-term impact of forest management and climate change across multiple scales. Realistically speaking, the biophysical perspective of this model is somewhat overly simplistic, as compared to many land surface models or dynamic vegetation models out there (e.g. OCHIDEE, CABLE-CNP etc.). While I agree with the authors that there may be a balance between realism and scalability, the authors failed to convince me that their model implementations are adequately enough to make some novelty claims that they stated in the manuscript (I detail these in the specific comments). For example, the authors claimed that their independent simulation of stomatal conductance and its linkage with plant hydraulic is a novelty of this study (e.g. P39, L3-4), but they never demonstrated how their simulated photosynthesis was coupled/decoupled with stomatal conductance, how water availability affected this relationship, and how well the model performed in relation to data. I suggest the authors to demonstrate the performance of this "novelty" in order to claim it.

Moreover, the other novelty that the authors claim was that, the model offers a large range of options of management. I think these management options are easily implementable in land surface models, and some may have already been implemented (detailed comments in specific comment). I think it's OK to claim these additional modelling implementations as novelty. However, I do feel that they haven't really demonstrated well enough how each, and the combination of these management options affect the simulated results. Their figure 3 and 4 for example, did not convince me that the new simulation really significantly improved the comparisons with observations. I suggest further sensitivity analyses on this point. Further along this line of thought, some representations of management and their effect on vegetation dynamics are supported with no literature evidence, and it seems that some are rather simplistic (i.e. without species/climate/soil –specific effects). I think this warrants some discussions.

Furthermore, I think there is a possible missed opportunity with regard to nutrient cycles. The authors claimed that they had some nutrient contents simulated, and leaf respiration depends on nitrogen. I understand that incorporating a full nutrient cycle may be beyond the current paper, but

the authors really didn't test how nutrient affect photosynthesis. I suggest the authors to justify the reasoning for not relating nutrient availability with photosynthesis (especially given the relationship between respiration and nitrogen), or make some simple tests to see how well/poorly their modelled nutrient availability was. This would point to interesting future research to improve the model, I think.

**Specific comments:**

Abstract: the $2^{nd}$ half of the abstract only described what the author have done – i.e. examines the sensitivity of the model, compares the model performance with observations. I would like to see some more explicit descriptions of the results of these actions.

P4, L2: This statement really depends on your definition of representations. Many land surface models did incorporate empirical relationships on management effect on soil and vegetation carbon. A recent literature is Felzer and Jiang (2018), who assessed the effect of different land uses on vegetation and soil carbon sequestration, including forest harvests. The relationships in their model are empirical, but so does some relationships described in this study.

Table 1: Any particular reason why atmospheric O2 concentration is an input in this model?

P8, L13: What depth is the reference depth? Can you specify?

P9, L8: So stomatal conductance is simulated independently of photosynthesis. Can you show, in your model evaluations, how photosynthesis and stomatal conductance is coupled/decoupled under different weather conditions? I think it's important for the readers to know the performance of these two fluxes, especially given the current way you represent these two inter-related fluxes.

P9, L9: Can you perform a sensitivity test on the time constant? This constant seems to potentially have a big effect determining your drought responses.

 Equation 13: How could one derive relationships from observations to drive your model? I can see many assumptions must have gone into the parameterization of this equation. How much confidence can we trust the model prediction, if these parameters were only empirically-determined/assumed?

P10, L25: You have maximum root depth as an input parameter, but how root depth changes with plant age?

Equation 24: You did not have nutrient effect on photosynthesis, but you included N effect on respiration? Can you justify the reason to not include N effect on photosynthesis then? That seems a missed opportunity given the current momentum in including nitrogen and phosphorus cycle processes in land surface models, which has been quite nicely reviewed in Achat et al. (2016) and evaluated in Fleischer et al. (2019).

P13, L2: "than" grammar issue?

P13, L6: Does allocation only respond to this water stress index and nothing else (e.g. nutrient, competition, phenology)? This could be quite an important weakness that needs further justification. Also, it seems that this water stress index only changes at annual timestep (P14, L5). Is this too coarse a resolution to simulate drought effect on growth and transpiration fluxes? The model certainly resolves energy, water and carbon budgets at hourly timestep, which implies that the model has the capacity to investigate detailed water-carbon relations under extreme conditions. But if the water stress index is only updated at annual timestep, I see little possibility for a realistic simulation of the diurnal and intra-annual variability in carbon-water coupling.

Figure 2: Allocation partitioning into different root components – how do you parameterize and evaluate this? For such a simple allocation scheme, maybe the authors want to justify the need for additional complexity in representing root dynamics. What additional insights do you gain by compartmenting roots into 4 categories?

Section 2.7: The representation of vegetation phenology includes very little mechanistic understanding – from what I can see, some part of the model only still uses date of year to change phenology. Maybe that's a point of future model improvement, but some acknowledge of the limitation may be needed.

P16, L21 – 22: From reading of this, it appears to me that you consider a tree dead once you can't close the carbon mass balance. Is this a realistic/safe assumption? The thing is, this assumption ignores the role of plant hydraulic and physiological traits in modulating plant responses to extreme conditions. I think some acknowledgement on the lack of process-based representation of tree mortality is needed here.

P16, L25: If I understand this correctly, here potentially coarse woody debris is added to soil pool?

Figure 3: Clearly the new prediction still can't capture the exact management effect, so what's the point of including these management options in your model? Yes the simulation is better matched with observation over the long-term, but the immediate impact should also be represented, I would argue.

Figure 4: Prediction not necessarily improved, is it?

P21, L10 – 14: these assumptions seem to be very arbitrary – no citations, and not species-specific.

Section 2.9.4: I don't think there is much mechanistic basis in these model implementations. And if you have nutrient concentration in leaf, it seems to be logical to include nutrient effect on plant photosynthesis, at least that's what the authors did for respiration. Some justifications are needed as to why the authors did not consider nutrient effect on photosynthesis. There are relationships available to do so (e.g. Walker et al., 2014).

The following section on sensitivity and parameterization test seems thorough, but I do note that the model was parameterized, so it's reasonable to see the model simulation matched with observations to some extent. I think it's more important to test the sensitivity of the assumptions that determine the $CO_2$, temperature, precipitation, etc. responses, which is a different suite of

sensitivity test. This different suite of sensitivity test would allow one to really entrust the model mechanisms to predict future climate change impact.

Table 6: why "continued"?

P39, L3-4: You haven't evaluated how photosynthesis couples/decouples with stomatal conductance under water stress. I think you need to demonstrate it before you call it a novelty of the paper.

P40, L5: there is an extra comma in the citation bracket.

Reference cited:

Achat DL, Augusto L, Gallet-Budynek A, Loustau D. (2016). Future challenges in coupled C–N–P cycle models for terrestrial ecosystems under global change: a review. Biogeochemistry 131: 173–202.

Felzer, B.S. & Jiang, M. (2018). Effect of land use and land cover change in context of growth enhancements in the United States since 1700: net source or sink? Journal of Geophysical research, Biogeosciences, 123, 3439-3457.

Fleischer, K., Rammig, A., De Kauwe, M.G. et al. (2019). Amazon forest response to CO2 fertilization dependent on plant phosphorus acquisition. Nat. Geosci. 12, 736–741.

Walker, A.P., Beckerman, A.P., Gu, L., Kattge, J., Cernusak, L.A., Domingues, T.F. *et al.* (2014). The relationship of leaf photosynthetic traits - $V_{cmax}$ and $J_{max}$ - to leaf nitrogen, leaf phosphorus, and specific leaf area: a meta-analysis and modeling study. *Ecology and Evolution*, 4, 3218-3235.

---

## Referee Comment (RC2) · Anonymous Referee #2 · 4 Jul 2020

The paper "Energy, water and carbon exchanges in managed forest ecosystems: description, sensitivity analysis and evaluation of the INRAE GO+ model, version 3.0" provides a comprehensive model description and results of additional model tests. It is a well written and well structured contribution and will be a solid base for future model applications and model users. One point of critique is a tendency for over-enthusiasm regarding the model. This does not support the understanding of the material and is not necessary in a modeling journal. For instance, in many instances the text reads (almost) as if trees were explicitly modelled ("... among individual trees ..."), which is

not the case. Similarly, phrases such as "each canopy layer..." are a bit misleading, as there is exactly one tree canopy layer (and an understorey layer) in the model. Also the introduction overstretches a bit (in my opinion), e.g. when hinting at the applicabilty at global level, while GO+ is (currently) applicable for a subset of the temperate biome (evenaged, mono-specific, managed). I would suggest to tone down the marketing language throughout the paper - the model is great and can speak for itself. Another major point is the "verifcation" section of the paper. In my point of view a technical test to show that a model is not leaking (or creating) energy or matter is such a basic step in model development that it does not merit a section on its own. Of course I might be wrong here - and reading about substantial differences (9%) is an indication that I am wrong here. Please explain better why such a test is (a) non-trivial and (b) avoid to create the impression that empirical data was involved in this test (in addition to initial conditions; see previous point). Some clarifications would also help the reader to understand the model. One is the spatial dimension which is never really mentioned; Is there a conceptual stand area for which the model designed? This may also help to better understand the management options available in the model. A second point is that the response functions (e.g., f_SW to calculate stomatal conductance) are not described (page 9). Later (page 24), it sounds as if those functions are specifically developed for each species, which would be an approach that is hard to scale up to many more species. Please include this information early in the manuscript.

A more minor point is that oftentimes abbreviations - particularly in figure captions - are not explained. Moreover, the paper is quite long already and the authors should consider shortening here and there (some suggestions in the details).

All in all I think the paper is an excellent piece of work that fits very well in the journal. The evaluation exercises are nicely executed and highlight results for different processes at different time scales. The publication of the model source is clean (github) and the use of publicly avaialable data sets (e.g., ISI-MIP) is another plus.

Details: page 3, line 4: better explain the mechanisms (sensible heat flux)

p3L18ff: you do not consider the life cycle of wood products. Your model can provide data that could allow such an analysis....

p6L1: "releasing model calculations...": nicely put!

p7l5: "latter" instead of "later"

p9l1: "is taken" ... "extended *the* use of Eqs"

p9l13: please be a bit mor

p10l5ff: the concept of dynamic layer dimensions is an interesting approach!

p11l19: extra space after "presented here"

p13: please make clear what "indivdual trees" are and what not

p16l17ff: you state that you do not model tree mortality; what about regeneration? Is this also limited to managed forests (with planting)? You could mention this here (growth/mortality/regeneration as the major demographic processes). And what is the "carbon balance of individual trees"???

p17l27: double ".."

p19l10: it is not clear what the "number of trees felled" means. How does the number of trees change in the model? By management (thinning, final cuts), right? How is that related to the mortality of trees? You see, I am confused.

p20 Fig5: please add explanations for the pools (DPM, RPM, ...) in the caption. There is a stray label ("L2") in the figure. Captions (e.g. "Long", "Beech" are way too big - or the other text labels are way too small.

p22l12: maybe use a different term instead of "verification". Maybe something like "testing conservation principles"?

p26/p27: in my opinion the discussion of results of the sensitivity analysis is too long and too verbose. It is meaningful to have such an analysis, but it is just limited what

can be derived from such a general +-10% approach.

p27Figure7: spell out abbreviations or refer to some table.

p30l1: How did you select the SDs of the parameters?

p33l30: "annual increase" instead of "time derivative".

p34Table 5: Can you add observed/predicted values here?

p35 Table 6: remove "continued"

p35l10: the sentence is duplicated ("The random errors...")

p36 Table 7: Is this analysis useful? It does basically say that the model is able to discern between summer and winter. I think this is a candidate for shortening the paper

p39l17: Remove the "the" after GO+

p39L22: confused again. How do you simulate the dynamics of size distribution??

p40L1: not true. There are gazillions of forest management practices that are not covered. What about single tree selection approaches or other spatial explicit small scale interventions? What about everything related to mixed forests?

p40l28: "Model performance" either remove or create paragraphs in the discussion

p45: Leaf area is m2/tree, not m-2/tree. same for BA. LAI is m2/m2, not m-2/m2. Stem volume is m3, stocking densitiy is stems... this page is a bit messed up.

---

## Author Comment (AC1) · 13 Aug 2020

documentclassarticle [utf8]inputenc graphicx

We are grateful to both reviewers for their careful reading and correction suggestions. We accounted for most of their comments for revising a new version of our manuscript. Both referees call for rewriting parts of the manuscript more rigorously regarding the novelties brought and originality of the GO+ model. In the revised version, we took this opportunity to document more accurately the originality of the model, minor our

over-enthusiastic statements and shorten some parts of the manuscript. We have organised our reply starting to address first the general comments of both referees and replying then to each specific comment.

**1 General comments of referees #1 and #2**

**1.1 Referee # 1**

Referee # 1 raised three main points.

- The linkage between soil water, plant size, plant, hydraulics and stomatal functioning and photosynthesis.

  We agree with the comments of both referees regarding the need for a clear demonstration of the model novelty in this area. Although kept simple, the explicit linkage between plant hydraulics and stomatal conductance in GO+ v3.0 is not implemented in most forest models where, e.g., the Ball-Berry's model is used instead (Guillemot et al. 2014) or a simpler LUE approach is used (Landsberg and Waring 1997) or time resolution is too large for calculating hourly values of the leaf water potential (Reyer et al. 2014). We simply meant that our stomatal and photosynthesis model is not based upon an optimality postulate and doe not link the stomatal conductance to photosynthesis. In the revised version, we propose to illustrate in a new figure (Figure 1) the effect of the leaf water potential on the stomatal conductance and photosynthetic $CO_2$ assimilation at a range of leaf-to-air water vapour pressure deficit, $CO_2$ concentration, and for different tree species. We agree our statement about the

originality of this formalism was a bit exaggerated and that our model is actually close from the original Jarvis's (1976). We rewrote accordingly the first part of the discussion. We think that demonstrating the performance of the model for specific processes would lengthen considerably the manuscript and be complex due to the paucity of appropriate data. The evaluation of the canopy fluxes of water vapour and CO2 modelled against flux data measured by eddy covariance makes realise how well / badly the model behaves for simulating photosynthesis and evapotranspiration.

• The effect of the management options implemented in the model on model results.

There is currently a vivid discussion about the role of forestry in the carbon emissions reduction policy of the countries that are signatories to the Paris Agreements. This makes crucial to describe realistically the impacts of forest operations on carbon pools in the soil, harvested products and biomass. However, the present and upcoming forest management strategies are poorly represented in most models. The main merit of our model is its versatility that allows to create, assemble and combine the forest operations currently included in management schemes of coppices, plantation forests (soil preparation, intensive harvesting of different tree parts, ground vegetation removal), as well as close-from-nature management schemes (thinnings from the top, from below or random, self thinning). Our model can describe a relatively vast range of sylviculture options for even-aged, monospecific forests. In addition, the soil carbon sub-model, adapted from Roth-C, accounts for the effects of soil operations (tillage, disking, moulding,...) and the improvement of the soil carbon dynamics illustrated in Fig 3 is a substantial progress. However, we agree that the model originality is lesser for assessing deforestation impacts which led us to remove the Figure 4 from the revised version. We add a sentence in the Discussion section making clear that
the version 3.0 is not including mixed or multilayered forests.

- The nutrient cycles and related impacts on photosynthesis and respiration.

  Thanks for the suggestion. The relationship between N and P leaf content, SLA and photosynthetic parameters is still an open area for modelling (Achat et al. 2016, Jiang et al. 2019). Meta-analysis comparing species (Walker et al. 2014) and previous modelling attempts developed in few global or forest models are of course inspiring but cannot be adapted straightforwardly to GO+. Based upon our recent synthesis (Achat et al. 2016)), a module describing the main processes of the nitrogen and phosphorus cycles in the soil-plant system is being developed for GO+ but goes beyond the scope of the v3.0 version .

  Hence, the simulations shown are all based assuming $V_{cmax}, J_{max}$ values at 25 °C are fixed because we have not yet evaluated the next nutrient version with data observed. Though, we implemented the nitrogen content effect on maintenance respiration in the V3.0 version shown because its ecophysiological basis is widely understood (Ryan, 1991) and thus easier to parameterise and make generic among species and plant parts. However, the concentration of nitrogen in each part of the plant is fixed, which would make it superfluous to further detail the photosynthesis - foliar nitrogen relationship.

**1.2 Referee # 2**

The referee # 2 pointed 1) the usefulness of the verification section and 2) the explanation of the stomatal response functions (Eq. 13).

- The "Verification" section complied with the requirement of the journal but we thank you for the suggestion of changing its title. We thought this section is useful for showing the closure of mass and energy balances on a 22 year-long time series. Here, the lack of closure of the energy balance (9 % ) is explained in this section by the approximation made about the atmospheric stability and the related underestimation of the convective heat fluxes by the model.

- We apologise because the sentence P9L10 " *The individual stomatal response functions used are allowed to vary according to the species considered*" is misleading and will be removed from the revised version. Actually, in the version 3.0 of the GO+ model, the response functions of stomatal conductance to shortwave radiation, leaf-to-air vapour pressure deficit, CO2 and leaf water potential are generic among species( see Figure 1), only their parameterisation is species specific. We add to Eq.13 the explicit description of the four response functions (previously shown in the parameterization section).

**2  Specific comments referee#1**

*Abstract: the 2nd half of the abstract only described what the author have done – i.e. examines the sensitivity of the model, compares the model performance with observations. I would like to see some more explicit descriptions of the results of these actions.* We changed the second part of the abstract.

*P4, L2: This statement really depends on your definition of representations. Many land surface models did incorporate empirical relationships on management effect on soil and vegetation carbon. A recent literature is Felzer and Jiang (2018), who assessed*

[Figure]

*the effect of different land uses on vegetation and soil carbon sequestration, including forest harvests. The relationships in their model are empirical, but so does some relationships described in this study.*

We partially agree. Feltzer and Jiang (added in references) nicely described the effects of land use change and timber harvesting at the country scale on the only biomass stocks and related litter input to the soil. They ignored the putative short term effects of soil preparation, timber logging and clearcutting on soil organic matter mineralisation and ground vegetation that are common in European forestry. In European forests, Naudts et al. (2016) and Luyssaert et al. (2018) accounted for the impacts of forestry on the soil carbon dynamics considering the only changes in litter inputs and exports. They do not estimate effects on organic carbon mineralisation and transformation dynamics due to soil preparation techniques. In the forests of Southern Europe, the removal of branches, stumps and foliage of trees and understory are also relatively common but not accounted for by most forest models.

*Table 1: Any particular reason why atmospheric O2 concentration is an input in this model? P8,* Michaelis-Menten coefficient of Rubisco is depending on CO2 and O2 concentrations.

*L13: What depth is the reference depth? Can you specify?* This is actually the soil depth. Corrected.

*P9, L8: So stomatal conductance is simulated independently of photosynthesis. Can you show, in your model evaluations, how photosynthesis and stomatal conductance*

*is coupled/decoupled under different weather conditions? I think it's important for the readers to know the performance of these two fluxes, especially given the current way you represent these two inter-related fluxes.*

We added a new figure deciphering the impact of leaf water potential and air water vapour pressure saturation deficit on the stomatal conductance and photosynthesis of the tree canopy.

*P9, L9: Can you perform a sensitivity test on the time constant? This constant seems to potentially have a big effect determining your drought responses.*

Yes we could but think it would lengthen the already long paper. We expect substantial effects on understorey exchanges where species with long response time, e..g. Fern species, are common. In tree species, the time constant is rarely exceeding 30-40 mn and has no impact on the model results at an hourly resolution.

*Equation 13: How could one derive relationships from observations to drive your model? I can see many assumptions must have gone into the parameterization of this equation. How much confidence can we trust the model prediction, if these parameters were only empirically- determined/assumed?*

We have explained this in the parameterisation section. The stomatal model equation is generic and may be parameterised at the canopy scale, using eddy covariance flux data, sap flow measurements, or upscaling gas exchange measurements performed with chambers. The new figure (Figure 1) will be inserted in the parameterisation section.

*P10, L25: You have maximum root depth as an input parameter, but how root depth changes with plant age?*

There is no changes in soil rooting depth with age in the V3.0 version. We agree that a relationship of the rooting depth with tree size is however missing for forest plantations and younger stands, where root expansion must be accounted for. We warn for this limitation in the discussion part of the version revised.

*Equation 24: You did not have nutrient effect on photosynthesis, but you included N effect on respiration? Can you justify the reason to not include N effect on photosynthesis then? That seems a missed opportunity given the current momentum in including nitrogen and phosphorus cycle processes in land surface models, which has been quite nicely reviewed in Achat et al. (2016) and evaluated in Fleischer et al. (2019).*

This is being developed in the next version of the GO+ model and will be evaluated using different forest experiments, including input from N fixing species, nitrogen deposition and fertilisation.

*P13, L2: "than" grammar issue?*

Yes, thanks, corrected.

*P13, L6: Does allocation only respond to this water stress index and nothing else (e.g. nutrient, competition, phenology)? This could be quite an important weakness that needs further justification. Also, it seems that this water stress index only changes at annual timestep (P14, L5). Is this too coarse a resolution to simulate drought effect*

*on growth and transpiration fluxes? The model certainly resolves energy, water and carbon budgets at hourly timestep, which implies that the model has the capacity to investigate detailed water-carbon relations under extreme conditions.*

Apart from the foliage, the growth of the other tree parts (stem, branches, root parts) is integrated over the year and updated annually. We used therefore an annual index for simulating the impact of water stress on the carbon allocation between above-ground and below-ground tree parts (Eq. 28).

*But if the water stress index is only updated at annual timestep, I see little possibility for a realistic simulation of the diurnal and intra-annual variability in carbon-water coupling.*

The impact of leaf water stress on transpiration and photosynthesis is implemented hourly through stomatal limitation (Eqs.13 and 16) and leaf growth (understorey). Hence, the water and carbon are coupled at a range of time frequencies:

- Hourly for most canopy processes:

  - changes in leaf temperature, and in turn respiration, controlled by stomatal closure
  - leaf water potential and VPD control of stomatal conductance and in turn the internal $CO_2$ concentration that affects photosynthetic assimilation of carbon

- Daily and yearly for growth, phenology and mortality processes.

These couplings are not covering all possible water stress effects on ecophysiological processes in trees, that are multiple. So, we do not pretend that the model is capturing

every ecophysiological process coupling carbon and water metabolism a tree.

*Figure 2: Allocation partitioning into different root components – how do you param-
eterize and evaluate this? For such a simple allocation scheme, maybe the authors
want to justify the need for additional complexity in representing root dynamics. What
additional insights do you gain by compartmenting roots into 4 categories?* This
parameterization is empirical and based upon allometric relationships detailed in
Achat et al. 2018. This is detailed in the Suppl Mat. Table S2 and Eqs. S34-S62.
This level of details is requested for calculating the biomass maintenance respiration.
It has not yet any other implication for the processes described in the V3.0. but, as
explained in section 2.9.4, this was needed also for calculating the nutrient export due
to harvests (Achat et al. 2018). Moreover, it will be used in the subsequent versions of
the model for a number of processes included in N and P nutrient cycles.

*Section 2.7: The representation of vegetation phenology includes very little mechanis-
tic understanding – from what I can see, some part of the model only still uses date
of year to change phenology. Maybe that's a point of future model improvement, but
some acknowledge of the limitation may be needed.*

As in most models, the bud burst, leaf unfolding and leaf duration (Fagus) are
depending on the accumulated temperature – as chilling or forcing temperatures
–, photoperiod (Quercus spp., data not shown) and accumulated radiation. Being
more mechanistic would lead to implement detailed physiological processes that are
complex and difficult to describe at the canopy scale. So, in this version, only the
needle cohort life duration is fixed. All other phenological variables are controlled by
temperature, radiation or water stress (Tables S3, S5).

*P16, L21 – 22: From reading of this, it appears to me that you consider a tree dead once you can't close the carbon mass balance. Is this a realistic/safe assumption? The thing is, this assumption ignores the role of plant hydraulic and physiological traits in modulating plant responses to extreme conditions. I think some acknowledgement on the lack of process-based representation of tree mortality is needed here.*

Yes, thanks it has been added in the discussion. We are aware of the numerous studies relating stomatal function to cavitation avoidance since, e.g., Jones and Sutherland (1991) until Choat et al. (2018). The implementation of variable resistances linked to cavitation and embolism of sapwood tissues and their repair would need to represent individual tree hydraulics that is not feasible with our model.

The negative carbon balance is produced when assimilation has been constrained by prolonged stomatal closure as a result of plant water stress. The stomatal response to leaf water potential is itself a consequence of plant hydraulics function (capacitance in tree and resistances in plant and soil). We assume that the GO+ hypothesis of the death of the tree of a negative carbon balance is a simple and direct modelling shortcut to summarize the impact of water stress in terms of tree mortality.

*P16, L25: If I understand this correctly, here potentially coarse woody debris is added to soil pool?*

Yes.

*Figure 3: Clearly the new prediction still can't capture the exact management effect, so what's the point of including these management options in your model? Yes the*

*simulation is better matched with observation over the long-term, but the immediate impact should also be represented, I would argue.*

Thanks. Our message here is essentially to show how GO+ captures the trajectory of soil carbon following a series of soil preparation operations. The example shown in Fig 3 are a unique time series of soil carbon content measured during the C. Jolivet Ph. D. thesis work in 2000. However, this time series is not fully consistent and should not have been used as such for the following reasons.

- First, the soil density used for calculating the observed values of C stocks was kept constant at a default value (1.13); it may have been changed during ploughing and disking but we have non information on this.

- Second, the slash and ground vegetation were initially accounted for until 1998 but were then piled in rows and left apart when ploughing (operation #5). They are not anymore accounted for in measurements until mid 1999. At that time, they were buried again in mineral soil during the subsequent disking (operation #4).

For consistency, we removed the observed C stock values measured without accounting for the slash and have changed the figure 3 accordingly (Figure 2).

*Figure 4: Prediction not necessarily improved, is it?*

See above. We removed it.

*P21, L10 – 14: these assumptions seem to be very arbitrary – no citations, and not species-specific.*

The fractions of understorey biomass destroyed by disking is the observed practices of the fast growing Pines plantations in Europe. We have added an explanation and reference. The calibration of the increase in the mineralisation and humification rates of the soil carbon are explained in section 2.9.1.

*Section 2.9.4: I don't think there is much mechanistic basis in these model implementations. And if you have nutrient concentration in leaf, it seems to be logical to include nutrient effect on plant photosynthesis, at least that's what the authors did for respiration. Some justifications are needed as to why the authors did not consider nutrient effect on photosynthesis. There are relationships available to do so (e.g. Walker et al., 2014).*

Yes thanks, this remark makes sense. We answered in the "General Comments" section, point #3.

*The following section on sensitivity and parameterization test seems thorough, but I do note that the model was parameterized, so it's reasonable to see the model simulation matched with observations to some extent. I think it's more important to test the sensitivity of the assumptions that determine the CO2, temperature, precipitation, etc. responses, which is a different suite of sensitivity test. This different suite of sensitivity test would allow one to really entrust the model mechanisms to predict future climate change impact.*

Yes we agree with this remark. However, we stress that, apart from the soil descriptive parameters (texture, soil depth), the model was not parameterized or calibrated on each site (L7 p35). Some parameters were obtained from the FR-LBr site and applied for all the othe stands shown in Figs. 13 (Pm-Vielle, Pm-Pomp21). Most model

parameters were calibrated from independent experiments not used in the evaluation tests.

Our sensitivity analysis is the basic analysis expected when describing the first version published of such model. Its objective is to show how the model responds to its parameters and climate and how this is affecting the model uncertainty. The sensitivity tests related to the shape of the response of each process to environmental variables would have lengthened substantially the manuscript. We plan to publish this analysis within a GO+ application paper (project $Forets - 21$ funded by the French Ministry of Agriculture and project $Biosylve$ funded by the French governmental agency *Ademe*).

*Table 6: why "continued"?*

A typo, thanks, removed.

*P39, L3-4: You haven't evaluated how photosynthesis couples/decouples with stomatal conductance under water stress. I think you need to demonstrate it before you call it a novelty of the paper.*

We agree and have added a new figure (see above and Figure 1)).

*P40, L5: there is an extra comma in the citation bracket.*

Thanks, corrected.

[Figure]

**3  Specific comments referee#2**

*page 3, line 4: better explain the mechanisms (sensible heat flux)* We add a short explanation regarding the effect in the boreal zone, the tropical impact of forests being already explained and referenced.

*p3L18ff: you do not consider the life cycle of wood products. Your model can provide data that could allow such an analysis....*

Agreed. We removed the sentence "It should be accounted for in forest models" that does not bring anything here.

*p6L1: "releasing model calculations...": nicely put!*

*p7l5: "latter" instead of "later"*

Thanks, corrected.

*p9l1: "is taken" ... " extended \*the\* use of Eqs" p9l13:*

Done.

*please be a bit mor ?*

In the revised version, the stomatal reduction functions have been developed in Eq.

13.

*p10l5ff: the concept of dynamic layer dimensions is an interesting approach!*

We agree, it is essential for including a groundwater table in the root zone.

*p11l19: extra space after "presented here"*

Done

*p13: please make clear what "individual trees" are and what not*

We changed the figure 2 legend and text.

*p16l17ff: you state that you do not model tree mortality; what about regeneration? Is this also limited to managed forests (with planting)? You could mention this here (growth/mortality/regeneration as the major demographic processes).*

Sound suggestion, thanks. We change this section into "Regeneration and Mortality" and explained how regeneration is prescribed in GO+.

*And what is the "carbon balance of individual trees"???* The carbon balance of individual trees is calculated at the end of the year as the difference between its annual

carbon gain, $GPP_i$, and annual carbon loss, $Rm_i + Rg_i$.

$$GPP_i - Rm_i - Rg_i \tag{1}$$

We have added a short explanation in the text.

*p17l27: double ".." Corrected.*

*p19l10: it is not clear what the "number of trees felled" means. How does the number of trees change in the model? By management (thinning, final cuts), right? How is that related to the mortality of trees? You see, I am confused.*

Sorry for that ! We changed this to "the number of cut or dead trees". The number of trees is continuously changing by mortality and regeneration. The mortality is mainly caused by the cutting of the stems harvested during thinnings or clearcutting and to a possible negative carbon balance of some individual trees.

*p20 Fig5: please add explanations for the pools (DPM, RPM, ...) in the caption. There is a stray label ("L2") in the figure. Captions (e.g. "Long", "Beech" are way too big - or the other text labels are way too small.*

The figure was redrawn.

*p22l12: maybe use a different term instead of "verification". Maybe something like "testing conservation principles"?*

See above.

*p26/p27: in my opinion the discussion of results of the sensitivity analysis is too long and too verbose. It is meaningful to have such an analysis, but it is just limited what can be derived from such a general +-10 % approach.*

We shortened the text by 25%.

*p27Figure7: spell out abbreviations or refer to some table.*

We refer now to the tables A1 and S1.

*p30l1: How did you select the SDs of the parameters?*

When available from their published references. Empirically in few cases

*p33l30: "annual increase" instead of "time derivative".*

Thanks, corrected.

*p34Table 5: Can you add observed/predicted values here?*

Yes, done.

*p35 Table 6: remove "continued"*

Done

*p35l10: the sentence is duplicated ("The random errors...")* Thanks, corrected.

*p36 Table 7: Is this analysis useful? It does basically say that the model is able to discern between summer and winter. I think this is a candidate for shortening the paper*

This analysis shows how the model error behave at a range of time frequencies. We agree that some frequencies reflect mainly obvious variations, e.g., the season, but find also useful to show how unsystematic errors tend to cancel out at low frequency.

*p39l17: Remove the "the" after GO+*

Done.

*p39L22: confused again. How do you simulate the dynamics of size distribution??*

The size distribution of the regenerating stand is prescribed. Each forest operations or self thinning is then selecting trees to be removed either from the top , the bottom or randomly. The natural death of trees provoked by a negative carbon balance may also affect either the bigger trees, the smaller trees or be random depending on the species-specific ratio between leaf area and respiring biomass.

*p40L1: not true. There are gazillions of forest management practices that are not covered. What about single tree selection approaches or other spatial explicit small scale interventions? What about everything related to mixed forests?*

See our reply in the General Comments section 1.

*p40l28: "Model performance"*

New paragraph created

*p45: Leaf area is m2/tree, not m-2/tree.*

Thanks, corrected.

*same for BA. LAI is m2/m2, not m-2/m2. Stem volume is m3, stocking density is stems... this page is a bit messed up.*

We agree, there was several typos here. We made all the corrections.

**4   References**

Choat, B., Jansen, S., Brodribb, T. J., Cochard, H., Delzon, S., Bhaskar, R., Bucci, S. J., Feild, T. S., Gleason, S. M., Hacke, U. G., Jacobsen, A. L., Lens, F., Maherali, H., Martínez-Vilalta, J., Mayr, S., Mencuccini, M., Mitchell, P. J., Nardini, A., Pittermann, J., Pratt, R. B., Sperry, J. S., Westoby, M., Wright, I. J., and Zanne, A. E.: Global convergence in the vulnerability of forests to drought, Nature, 491, 752-755, 2012.

Felzer, B. S. and Jiang, M.: Effect of Land Use and Land Cover Change in Context of Growth Enhancements in the United States Since 1700: Net Source or Sink?, Journal of Geophysical Research: Biogeosciences, 123, 3439-3457, 2018.

Jiang, M., Caldararu, S., Zaehle, S., Ellsworth, D. S., and Medlyn, B. E.: Towards a more physiological representation of vegetation phosphorus processes in land surface models, New Phytol, 222, 1223-1229, 2019.

Jones, H. G. and Sutherland, R. A.: Stomatal control of xylem embolism, Plant, Cell and Environment, 14, 607-612, 1991.

figure1.png

**Fig. 1.** Modelled response of the stomatal conductance (left) and light-saturated photosynthesis (right) to decreasing leaf water potential at three levels of air water vapour saturation deficit, $\delta e_w$: (A) 500 Pa, (B) 1500 Pa, (C) 3000 Pa. The response curves delineates the range of response of four tree species for atmospheric concentrations in $CO_2$ varying between 410 (full line) to 820 ppm (dashed line).

figure2.png

**Fig. 2.** (Revised version of the original Figure 3 of the main manuscript). Changes in the soil organic carbon stock during the regeneration phase following a clear-cut of a maritime pine stand as simulated by the GO+ model with and without adaptation for soil preparation (full and dotted lines respectively) and measured in the field (grey dots). Data taken from Jolivet (2000). The numbers inset in black dots refer to the forest operation. 1: Clear-cutting and logging; 2: Heavy disking; 3: Stump removal; 4: Cover crop; 5: Tillage; 6: Vegetation crushing.